# p50 mono-ubiquitination and interaction with BARD1 regulates cell cycle progression and maintains genome stability

Longtao Wu [1], Clayton D. Crawley[1], Andrea Garofalo[1], Jackie W. Nichols[1], Paige-Ashley Campbell[1], Galina F. Khramtsova [2], Olufunmilayo I. Olopade [2], Ralph R. Weichselbaum[3] & Bakhtiar Yamini [1✉]

p50, the mature product of *NFKB1*, is constitutively produced from its precursor, p105. Here, we identify BARD1 as a p50-interacting factor. p50 directly associates with the BARD1 BRCT domains via a C-terminal phospho-serine motif. This interaction is induced by ATR and results in mono-ubiquitination of p50 by the BARD1/BRCA1 complex. During the cell cycle, p50 is mono-ubiquitinated in S phase and loss of this post-translational modification increases S phase progression and chromosomal breakage. Genome-wide studies reveal a substantial decrease in p50 chromatin enrichment in S phase and Cycln E is identified as a factor regulated by p50 during the G1 to S transition. Functionally, interaction with BARD1 promotes p50 protein stability and consistent with this, in human cancer specimens, low nuclear BARD1 protein strongly correlates with low nuclear p50. These data indicate that p50 mono-ubiquitination by BARD1/BRCA1 during the cell cycle regulates S phase progression to maintain genome integrity.

[1] Department of Surgery, Section of Neurosurgery, The University of Chicago, Chicago, IL 60637, USA. [2] Department of Medicine, The University of Chicago, Chicago, IL 60637, USA. [3] Department of Radiation and Cellular Oncology, and The Ludwig Center for Metastasis Research, The University of Chicago, Chicago, IL 60637, USA. ✉email: byamini@surgery.bsd.uchicago.edu

Nuclear factor-kappa B (NF-κB) plays a complex role in genome maintenance and carcinogenesis. While the oncogenic properties of this transcription factor are well recognized[1], NF-κB also promotes tumor suppression[2]. This diversity is in large part attributable to the fact that NF-κB is comprised of multiple subunits, p50 (NF-κB1), p52 (NF-κB2), p65 (RELA), RELB, and CREL that dimerize in a variety of combinations[3]. While overexpression and phosphorylation of p65 has become a virtual hallmark of aggressive cancer and inflammation, NF-κB1/p50 is more commonly associated with tumor suppression[4–6].

NFKB1 gives rise to two proteins, p105 and p50. The mature product, p50, dimerizes and binds DNA, whereas p105 is cytoplasmic and behaves as an inhibitor κB (IκB) protein[7]. p50 is formed following ubiquitin-mediated processing of p105[8,9]. A central feature of p50 is that it is constitutively produced[10]. Moreover, although NF-κB dimers are generally retained in the cytoplasm, p50 has a propensity to distribute to the nucleus[11]. Consistent with this, even in unstimulated cells there is a significant amount of basal, DNA-bound p50. p50 dimers are generally thought to inhibit gene expression, yet they can also activate transcription either via interaction with co-regulators or simply as a result of loss of basal chromatin binding. p50, like other NF-κB subunits, modulates the response to DNA damage[7], and previous studies indicate that p50 is phosphorylated in response to ataxia telangiectasia and Rad3-related (ATR)-dependent signaling[12,13].

BRCA1-associated RING domain-1 (BARD1) is an essential protein best known as the main binding partner of BRCA1. Functionally, BARD1 dimerizes with BRCA1 and together the complex acts as an E3 ubiquitin ligase inducing mono- and poly-ubiquitination[14,15]. The BARD1/BRCA1 complex has known functions in homology-directed DNA repair (HDR), cell cycle regulation, and tumor suppression[16–18]. Many cancer-associated BARD1 missense mutations localize to its C-terminal BRCT domains[19,20]. Given that these domains are important in promoting phospho–protein interaction[21,22], factors that interact with them likely play a role in maintaining genome stability and potentially promoting tumor suppression.

In the current study, we identify BARD1 as a p50-interacting factor. p50 directly binds the BARD1 BRCT domains via a phospho-serine-binding motif. This interaction enables BARD1, with BRCA1, to mono-ubiquitinate p50 at two C-terminal lysines, a modification that occurs during S phase of the cell cycle. Functionally, loss of p50 mono-ubiquitination leads to destabilization of p50 protein resulting in deregulation of S phase and chromosomal breakage. These results, in combination with the strong correlation between nuclear p50 and BARD1 in clinical cancer specimens, suggest that the BARD1-p50 interaction plays a central role in the tumor suppressive effects of these proteins.

## Results

### p50 interacts with BARD1 BRCT domains in response to ATR.
Phosphorylation of p50 S329 (referred to here as S328 based on UniProt isoform 1: P19838-1) was previously shown to be required for genome stability[13]. To identify proteins that modulate this response, we used affinity purification of HA-tagged wild type p50 (p50$^{wt}$) or an S328A mutant (p50$^{S328A}$). Following immuno-purification and sodium dodecyl sulfate–polyacrylamide gel electrophoresis (SDS–PAGE), a unique band was found (Fig. 1a). Liquid chromatography–mass spectrometry (LC-MS/MS) analysis of this band identified BARD1 as a p50-interacting peptide (Supplementary Fig. 1a and Supplementary Table 1). p65 was also identified as an interacting factor, providing validation that the data represented factors associated with p50. The

interaction of p50 with BARD1 was verified by reciprocal co-immunoprecipitation (Co-IP) following overexpression of both proteins (Fig. 1b). Also, endogenous association of p50 and BARD1 was demonstrated in several cell lines, including primary mouse embryonic fibroblasts (MEFs), HeLa cells and MCF-7 breast cancer cells (Fig. 1c and Supplementary Fig. 1b). We then examined the interaction of BARD1 with p50 following knockdown of BRCA1. Even though BARD1 protein was substantially reduced with BRCA1 depletion, likely due to BARD1 destabilization[14], BARD1 still interacted with p50 (Supplementary Fig. 1c), suggesting that BARD1 and p50 interact independently of BRCA1. In addition, knockdown of BCL3, a factor previously reported to bridge the interaction of p50 and BARD1[23], also had no effect on the association of BARD1 with p50 (Supplementary Fig. 1d).

As ATR induces p50 S328 phosphorylation[12,13], we examined whether ATR altered the p50/BARD1 interaction. Treatment of cells with hydroxyurea (HU), a ribonucleotide reductase inhibitor that induces ATR and replication stress (RS), increased the association of p50 with BARD1 (Fig. 1d). To more specifically examine the role of ATR in this response, we used a previously described tamoxifen (TAM)-inducible vector to activate ATR[24]. This system involves stable expression of a construct bearing the activation domain of TopBP1, a protein that can independently induce ATR[25], fused to the estrogen receptor (TopBP1$^{ER}$) (Fig. 1e). Stimulation of cells expressing this construct with TAM activates ATR without forming DNA double-strand breaks (DSBs)[24]. Addition of TAM to cells expressing TopBP1$^{ER}$, or a GFP$^{ER}$ control, increased the association of p50 with BARD1 only in TopBP1$^{ER}$-expressing cells (Fig. 1e and Supplementary Fig. 1e). These results indicate that ATR induces the interaction of BARD1 and p50.

BARD1 has several structural motifs, including an N-terminal RING domain, three tandem ankyrin (ANK) repeats, and two C-terminal BRCT domains (Fig. 1f). To identify which BARD1 domain was required for interaction with p50, we incorporated previously described deletion constructs that remove specific BARD1 motifs[26]. p50 only interacted with BARD1 constructs that retain the BRCT domains (Fig. 1g). Consistent with this, the BARD1 peptide sequence that interacted with p50 on the initial MS/MS analysis mapped to amino acids 620-637 within the first BRCT domain (Supplementary Fig. 1a). Finally, we examined this interaction using purified proteins in a cell free system. 6His-p50 interacted with GST-BRCT (containing GST fused to the BARD1 BRCT domains) but not with GST alone, demonstrating the direct interaction of these two proteins (Fig. 1h). These finding indicate that ATR induces the interaction of p50 with the BARD1 BRCT domains.

### BARD1 interacts with a conserved p50 phospho-serine motif.
Tandem BRCT domains interact with phospho-serine peptides[21], and BRCT domains from different proteins have significantly different motif specificities. A potential BARD1 BRCT phospho-serine-binding motif was previously identified with the general sequence of pS-[DE]-[DE]-E[27]. We searched p50 for a potential BRCT-binding motif and identified a sequence based on Serine 337 (Fig. 2a). This sequence, SDLE, only differs from the previously identified BARD1 BRCT-binding motif at the +2 position and is highly evolutionarily conserved. To examine the role of S337 in the interaction of BARD1 with p50, we mutated this residue to alanine or the phospho-mimetic, aspartic acid. While p50$^{S337A}$ did not interact with BARD1, p50$^{S337D}$ interacted strongly (Fig. 2b).

The importance of S337 and the ability of ATR to induce the interaction of BARD1 with p50 suggested that ATR promoted

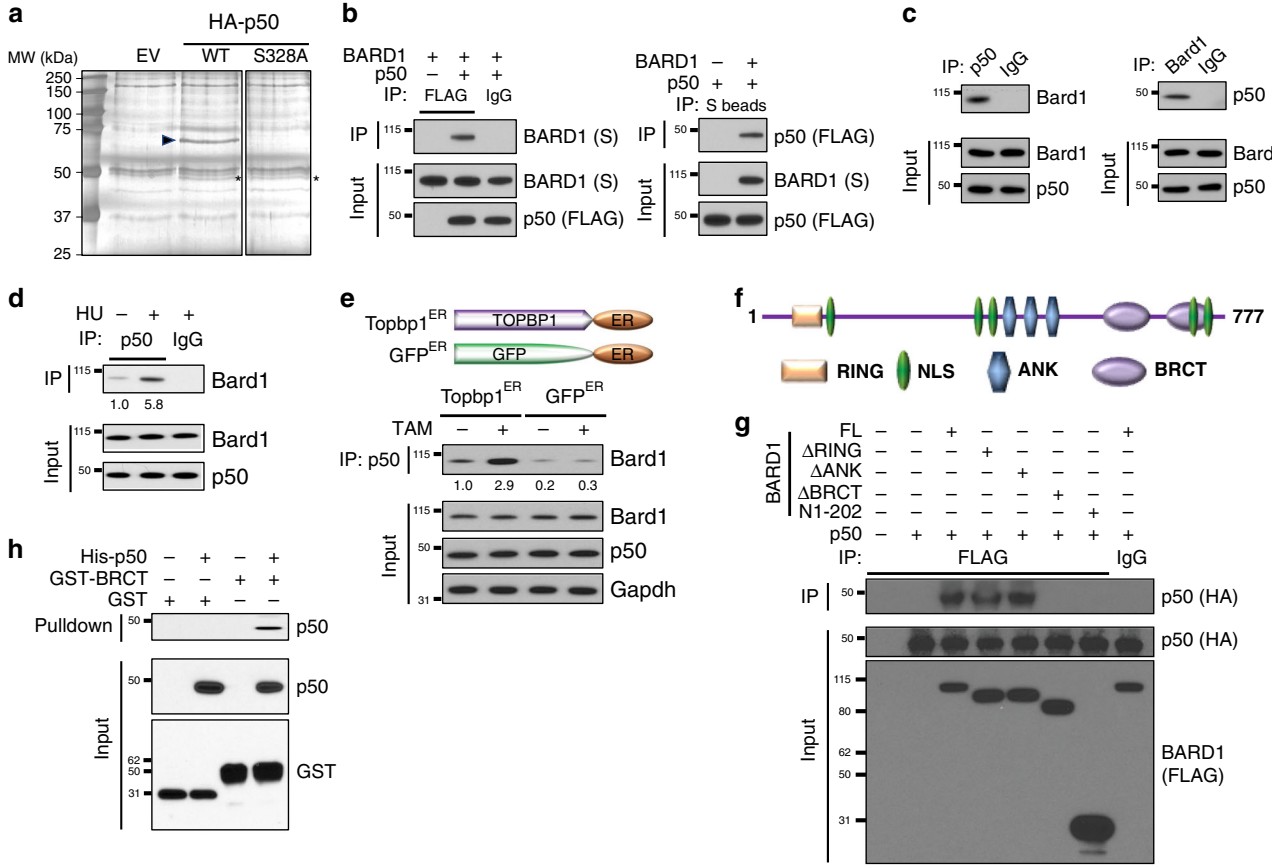

**Fig. 1 p50 binds the BARD1 BRCT domains. a** Silver stained gel of nuclear extract from 293T cells transfected with empty vector (EV), HA-p50[wt] or HA-p50[S328A]. Immunoprecipitation (IP) was performed with anti-HA. Asterisk indicates exogenous p50. Arrowhead, band analyzed by MS/MS. **b** Reciprocal Co-IP in 293T cells expressing S-tag BARD1 and FLAG-p50. IP with anti-FLAG (left) or S-agarose (right). Immunoblot (IB) with the indicated antibody. **c** Co-IP in primary wild type (WT) MEFs following IP of endogenous protein and IB with indicated antibodies. **d** Co-IP in WT MEFs treated with vehicle or HU (2 mM, 4 h). IP with anti-p50 and IB with anti-Bard1. **e** WT MEFs expressing TopBP1[ER] or GFP[ER] were treated with tamoxifen (TAM, 4 h). IP with anti-p50 and IB with anti-Bard1 or indicated antibody. Inset: schematic of TopBP1[ER] or GFP[ER]. **f** Schematic representation of BARD1 (ANK, ankyrin repeat). **g** Co-IP in 293T cells transfected with FLAG-BARD1 deletion constructs (FL, full length; N1-202, amino acids 1-202) and HA-p50. IP performed with anti-FLAG and IB with anti-HA. **h** Nickel column purification of 6xHis-p50 incubated with GST-BRCT or GST alone. Eluted proteins were examined with anti-p50 antibody. Input sample was probed with anti-p50 or anti-GST. All blots are representative of at least two biologically independent experiments. Analysis of fold-change normalized to control lane shown below IB where indicated.

phosphorylation of this residue. To examine S337 phosphorylation, we raised a phospho-S337 antibody and verified its specificity by phospho-peptide competition (Supplementary Fig. 2a). Activation of ATR in TopBP1[ER]-expressing cells with TAM or with an RS-inducing agent, induced S337 phosphorylation (Supplementary Fig. 2a). Time course studies demonstrated that TAM-induced S337 phosphorylation within 1 h that became maximal after 4 h (Supplementary Fig. 2b). To further examine S337 phosphorylation in response to ATR, we treated TopBP1[ER], or GFP[ER], cells expressing either p50[wt] or p50[S337A] with TAM. While an increase in S337 phosphorylation was seen in TopBP1[ER] cells expressing p50[wt], no phosphorylation was present at baseline or following TAM treatment in cells expressing p50[S337A] (Fig. 2c).

An association between p50 S337 and ATR signaling has not previously been reported. CHK1 is activated by ATR[28] and S337 is found within a minimal CHK1 consensus phosphorylation motif (RKKS, Fig. 2a)[29]. Knockdown of *CHK1* in cells expressing TopBP1[ER] blocked TAM-induced S337 phosphorylation (Fig. 2d). To examine whether CHK1 could directly phosphorylate S337, in vitro kinase assay was performed using purified p50 and active recombinant CHK1. Whereas CHK1 phosphorylated p50[wt], mutation of S337 to alanine blocked this effect (Fig. 2e), a

finding also seen with mutation of S328 as previously described[12]. Notably, S337 was phosphorylated in response to HU even when S328 was mutated (Supplementary Fig. 2c). Consistent with the role of CHK1 in p50 phosphorylation, TAM induced the interaction of endogenous p50 and CHK1 in cells expressing TopBP1[ER] but not GFP[ER] (Fig. 2f). Also, knockdown of *CHK1* blocked the interaction of BARD1 with p50 (Fig. 2g). These findings indicate that CHK1-dependent p50 phosphorylation promotes the association of BARD1 with p50. This finding was further supported by the observation that phosphatase treatment blocked the interaction of p50 with BARD1 in human and mouse cells (Fig. 2h and Supplementary Fig. 2d).

To examine whether a motif is required for the interaction of p50 with BARD1, we constructed a p50 "motif-mutant" in which S337 was retained but both D338 and E340 were mutated to alanine (p50[DE2A]) (Fig. 2a). Mutation of these two residues blocked the interaction of p50 with BARD1 (Fig. 2i). Moreover, in cell free studies with purified proteins, we found that unlike 6His-p50[wt], 6His-p50[DE2A] did not bind the BARD1 BRCT domains (GST-BRCT) (Fig. 2j). As a final specificity control of the phospho-dependent interaction, we mutated critical residues in the BARD1 BRCT phospho–protein interaction pockets (S575 and T617 in $P_1$ and H686 in $P_2$)[30]. Whereas p50 bound wild type

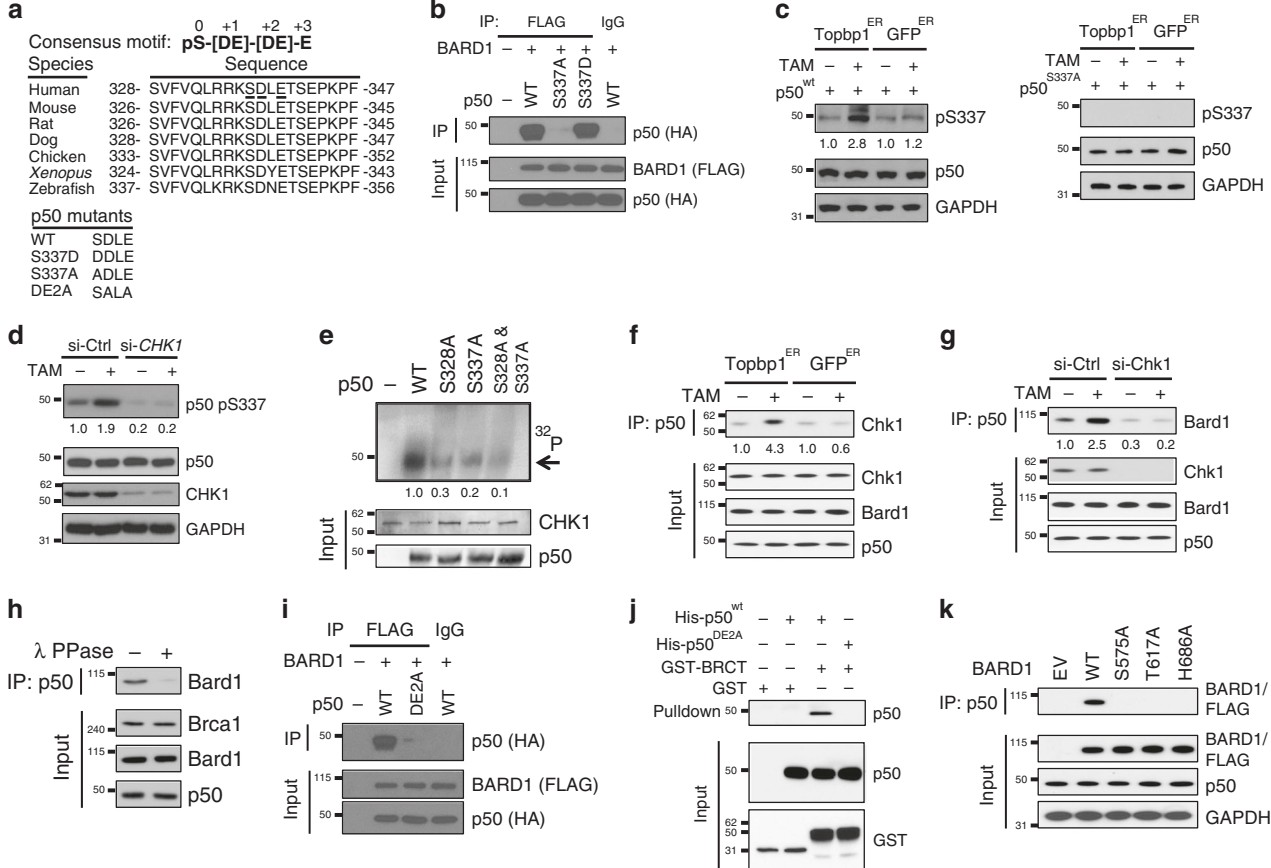

**Fig. 2 BARD1 interacts with a conserved p50 phospho-serine motif. a** Putative BARD1 BRCT-binding motif. Motif in human p50 is underlined. Sequence of p50 mutants (lower). **b** 293T cells expressing FLAG-BARD1 and the indicated HA-p50 construct. IP and IB performed with the indicated antibodies. **c** 293T cells expressing TopBP1$^{ER}$ or GFP$^{ER}$ transfected with p50$^{wt}$ (left) or p50$^{S337A}$ (right) and treated with vehicle or TAM (4 h). IP with anti-HA and IB with anti-phospho-p50-S337 antibody. **d** 293T cells expressing TopBP1$^{ER}$ were transfected with si-CHK1 or si-control and treated with vehicle or TAM. IB was performed with the indicated antibody. **e** Kinase assay with bacterially expressed p50$^{wt}$ or the indicated p50 mutant and recombinant CHK1. Autoradiogram ($^{32}$P) and IB with indicated antibody of the same membrane. Arrow, phosphorylated p50. **f** Co-IP in primary WT MEFs expressing TopBP1$^{ER}$ or GFP$^{ER}$ treated with vehicle or TAM (4 h). IP with anti-p50 and IB with the indicated antibody. **g** WT MEFs expressing TopBP1$^{ER}$ transfected with si-CHK1 were treated with TAM (4 h). IP with anti-p50 and IB with the indicated antibody. **h** Co-IP in WT MEFs following endogenous IP with anti-p50 and IB with anti-Bard1 or the indicated antibody. Lysates were treated with lambda phosphatase (λ PPase) or vehicle as indicated. **i** Co-IP in 293T cells expressing FLAG-BARD1 and the indicated HA-p50 construct. IP with anti-FLAG and IB with anti-HA antibody. **j** Nickel column purification of 6His-p50$^{wt}$ or 6His-p50$^{DE2A}$ incubated with GST-BRCT or GST alone. IB of eluted protein performed with anti-p50 antibody. **k** Co-IP in 293T cells transfected with empty vector (EV) or the indicated FLAG-BARD1 construct. IP was performed with anti-p50 and IB with anti-FLAG. All blots are representative of at least two biologically independent experiments. Analysis of fold-change normalized to control lane shown below IB where indicated.

(WT) BARD1, p50 did not bind BARD1 mutated at BRCT residues critical for phospho–protein interaction (Fig. 2k). Altogether, these findings indicate that phosphorylation of p50 S337 by CHK1 activates a BARD1 BRCT-binding motif enabling the interaction of p50 with BARD1.

**BARD1 promotes p50 mono-ubiquitination.** The BARD1/BRCA1 complex has ubiquitin ligase activity. We therefore examined p50 ubiquitination and found that treatment of cells expressing TopBP1$^{ER}$ with TAM-induced p50 ubiquitination within 2 h (Fig. 3a and Supplementary Fig. 3a). While some increase in the poly-ubiquitin smear was seen, the most prominent finding was an increase in a discreet band consistent with mono-ubiquitin addition. Similarly, treatment with HU also induced ubiquitination of p50 (Supplementary Fig. 3b). Knockdown of *BARD1* blocked basal and TAM-induced p50 mono-ubiquitination (Fig. 3b). To further examine the role of BARD1, we obtained *Bard1*-null mouse mammary carcinoma cells[26].

Although no p50 ubiquitination was seen in the absence of Bard1, expression of BARD1 resulted in the appearance of a prominent band consistent with mono-ubiquitinated p50 (Fig. 3c). To examine whether the discreet band was in fact mono-ubiquitinated p50, we used a mutant ubiquitin construct with no lysines (0KUb) that cannot undergo poly-ubiquitination. In the presence of 0KUb, expression of BARD1 still resulted in p50 ubiquitination (Fig. 3d) confirming that p50 is mono-ubiquitinated. As a specificity control, we used a G75A/G76A ubiquitin mutant (Ub GG-AA) that cannot be covalently conjugated to a substrate[31]. p50 was not ubiquitinated in the presence of this mutant (Supplementary Fig. 3c).

As BARD1 interacted with a p50 phospho-motif, we examined the role of this motif in p50 mono-ubiquitination. Using *Bard1*-null cells, we found that in the presence of BARD1, mutation of p50 S337 obliterated the mono-ubiquitin band without altering poly-ubiquitination (Fig. 3e). In these cells, expression of BARD1 also promoted mono-ubiquitination of p50$^{wt}$ but not p50$^{DE2A}$ (Fig. 3f). Moreover, while treatment with HU increased

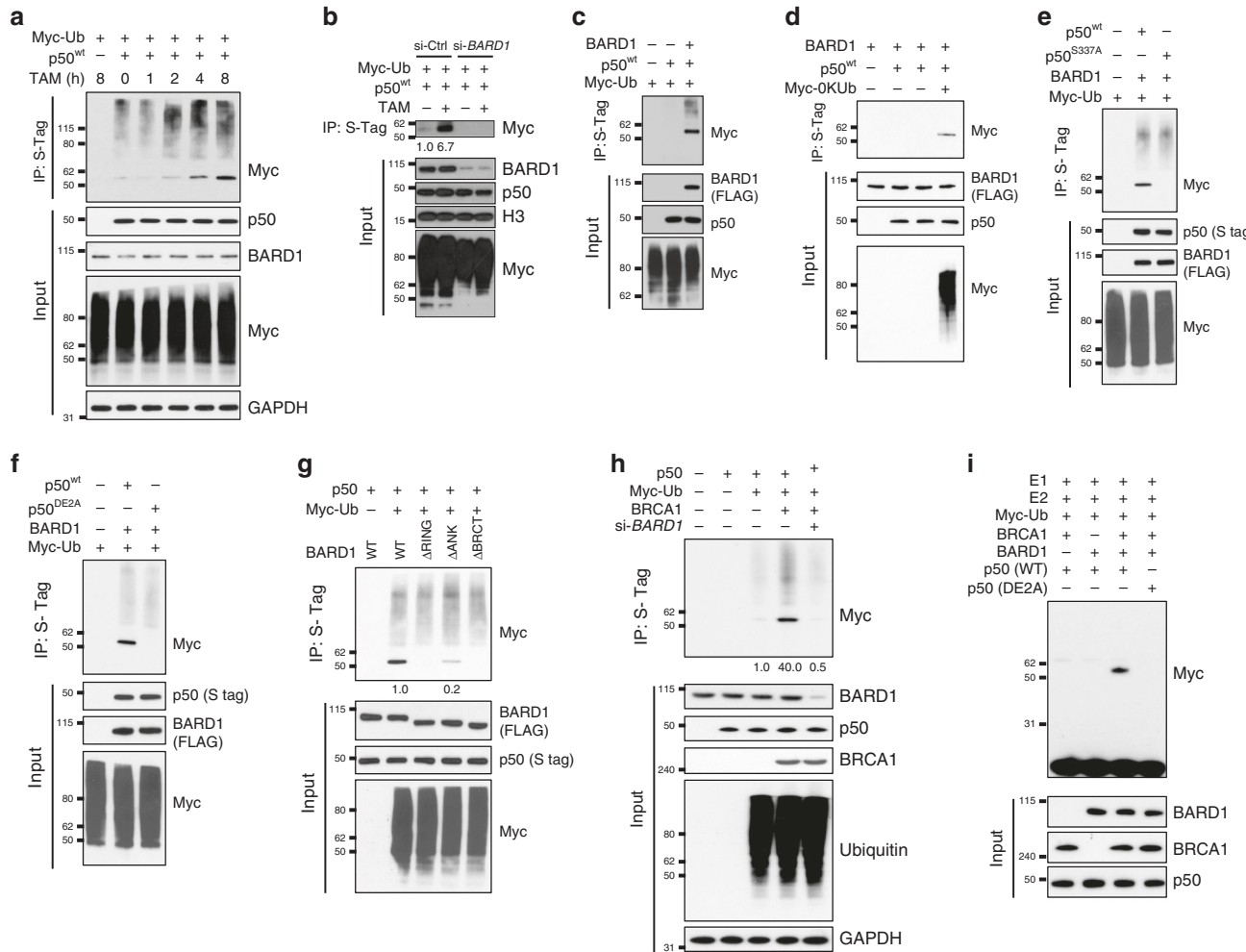

**Fig. 3 BARD1 binding promotes p50 mono-ubiquitination. a** Immunoblot (IB) in 293T cells expressing TopBP1[ER] transfected with Myc-ubiquitin (Myc-Ub) and S-tag p50[wt] treated with tamoxifen (TAM) for the indicated time. IP performed with S-agarose and IB with anti-Myc antibody. **b** Cells transfected as in A and si-*BARD1* or si-control treated with TAM (4 h). Nuclear fractions were isolated and IP and IB performed as in A. H3, Histone H3. **c** *Bard1*-null mouse cells transfected with Myc-Ub, S-tag p50[wt] and FLAG-BARD1. IP with S-agarose and IB with anti-Myc antibody. **d** *Bard1*-null cells transfected with FLAG-BARD1, S-tag p50[wt], and Myc-0KUb. IP with S-agarose and IB with anti-Myc antibody. **e** *Bard1*-null cells transfected with FLAG-BARD1, Myc-Ub, S-tag p50[wt], or p50[S337A]. IP performed with S-agarose and IB with anti-Myc antibody. **f** *Bard1*-null cells transfected with FLAG-BARD1, Myc-Ub, and S-tag p50[wt] or p50[DE2A]. IP with S-agarose and IB with anti-Myc antibody. **g** 293T cells transfected with Myc-Ub, S-tag p50[wt], and FLAG-BARD1 wild type or the indicated BARD1 deletion mutants. IP with S-agarose and IB with anti-Myc antibody. **h** HCC1937 *BRCA1* mutant breast cancer cells were transfected with S-tag p50[wt], Myc-Ub, HA-BRCA1, and si-*BARD1* as indicated. IP with S-agarose and IB with anti-Myc antibody. **i** In vitro ubiquitination using the indicated 6HIS-p50 protein incubated with GST-FL-BARD1, recombinant BRCA1, and Myc-Ub in the presence of UBE1 (E1) and Ubc5c (E2). IB performed with the indicated antibodies. All blots are representative of at least two biologically independent experiments. Analysis of fold-change normalized to control lane shown below IB where indicated.

mono-ubiquitination of p50[wt] without altering its basal poly-ubiquitination, p50[S337A] was neither mono-ubiquitinated at baseline or following treatment (Supplementary Fig. 3d). These results indicated that p50 interaction with BARD1 is necessary for p50 mono-ubiquitination not poly-ubiquitination.

Next, we examined the BARD1 domains required for p50 mono-ubiquitination. Deletion of the BARD1 BRCT domains blocked p50 mono-ubiquitination, as did loss of the BARD1 RING domain (Fig. 3g). Interestingly, although the ANK repeat region did not mediate BARD1 binding to p50 (Fig. 1g), loss of this region reduced p50 mono-ubiquitination (Fig. 3g), suggesting that the ANK repeats contributed to p50 mono-ubiquitination. We also examined the role of BRCA1 in modulating p50 ubiquitination. Using the BRCA1-mutated cell line, HCC1937[32], we found no significant p50 mono-ubiquitination in the absence of BRCA1 (Fig. 3h). In these cells, expression of BRCA1 led to

mono-ubiquitination that was lost with knockdown of *BARD1* (Fig. 3h). To further delineate the requirement of BRCA1, we examined this pathway in MEFs following knockdown of *Brca1* or *Bard1*. Importantly, to counter the inherent destabilization of these proteins by depletion of the other, we also over-expressed each factor. In this setting, depletion of either *Bard1* or *Brca1* blocked p50 mono-ubiquitination (Supplementary Fig. 3e). Finally, to examine whether the BARD1/BRCA1 complex directly ubiquitinates p50, we examined in vitro ubiquitination using purified proteins. In this system, p50[wt] but not p50[DE2A] was mono-ubiquitinated only in the presence of both BARD1 and BRCA1 (Fig. 3i). These results indicate that BARD1 directly binds p50 and together with BRCA1 mono-ubiquitinates p50.

**p50 is mono-ubiquitinated at its C-terminal.** To identify the p50 residue mono-ubiquitinated, we constructed a series of p50

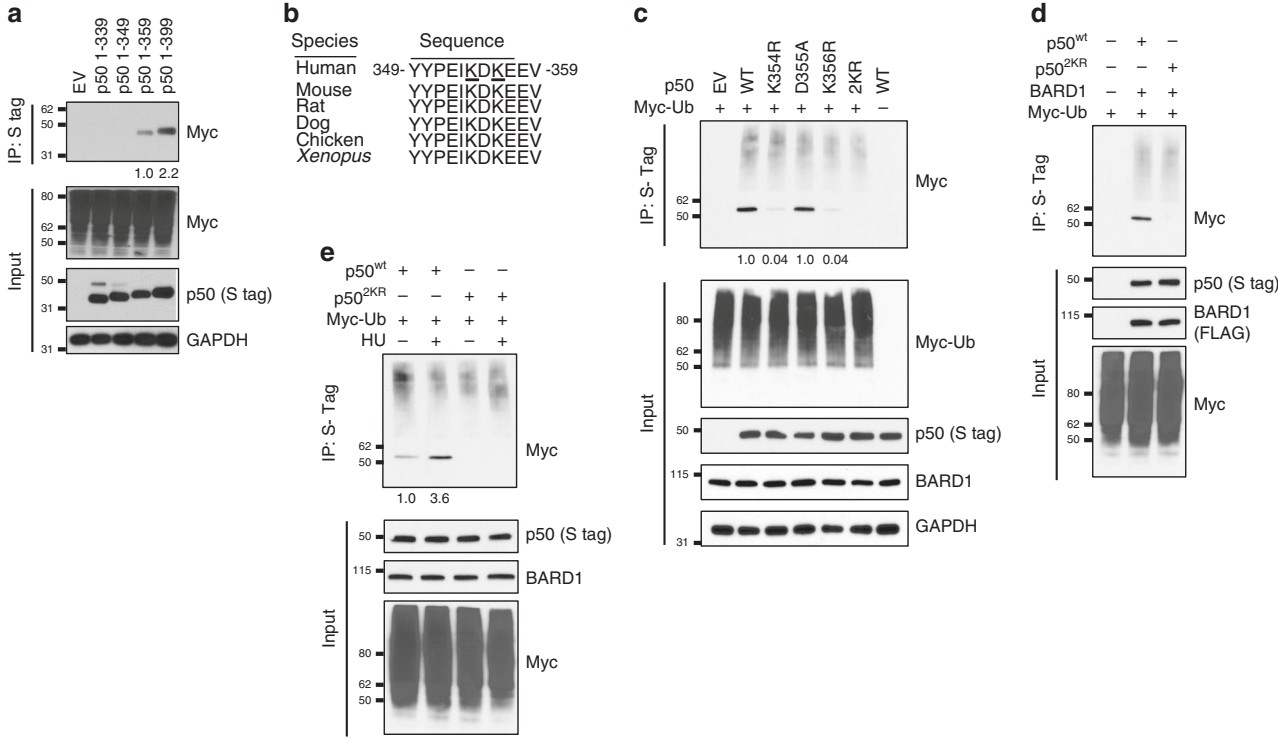

**Fig. 4 p50 is mono-ubiquitinated at K354 and/or K356. a** Co-IP in 293T cells transfected with Myc-Ub and the indicated S-tag p50 deletion construct. IP with S-agarose and IB with anti-Myc antibody. **b** Alignment of p50 sequences. Potential human mono-Ub sites underlined. **c** 293T cells transfected with Myc-Ub, S-tag p50$^{wt}$, or the indicated p50 mutant. 2KR, K354R/K356R. IP with S-agarose and IB with anti-Myc antibody. **d** *Bard1*-null cells transfected with FLAG-BARD1, Myc-Ub, S-tag p50$^{wt}$, or p50$^{2KR}$. IP with S-agarose and IB with anti-Myc antibody. **e** 293T cells transfected with Myc-Ub, S-tag p50$^{wt}$ or p50$^{2KR}$, and treated with HU (2 mM, 4 h). IP with S-agarose and IB with anti-Myc antibody. All blots are representative of at least two biologically independent experiments. Analysis of fold-change normalized to control lane shown below IB where indicated.

truncation mutants. Whereas p50 containing 319 amino acids did not bind BARD1, when at least 339 amino acids were present BARD1 binding was maintained (Supplementary Fig. 3f). Also, while p50 containing 359 residues was mono-ubiquitinated, p50 1–349 was not (Fig. 4a) suggesting that mono-ubiquitination occurred between residues 349–359. These ten amino acids are strictly evolutionarily conserved and contain two lysines, K354 and K356 (human) (Fig. 4b). We mutated these lysines, and the intervening aspartic acid, and examined p50 ubiquitination. Mutation of either lysine alone substantially decreased the mono-ubiquitin band and double mutation, p50 K354R/K356R (2KR), obliterated mono-ubiquitination with minimal effect on poly-ubiquitination (Fig. 4c). Notably, mutation of the intervening residue, p50$^{D355A}$, had no effect on mono-ubiquitin addition supporting the specificity of the lysines themselves and not the entire region for p50 ubiquitination. Importantly, these ubiquitin-site mutants retain the ability to bind BARD1 (Supplementary Fig. 3g).

To further examine these two residues, we expressed p50$^{wt}$ or p50$^{2KR}$ together with BARD1 in *Bard1*-null cells. In this system, p50$^{wt}$ was mono-ubiquitinated in the presence of BARD1 while p50$^{2KR}$ was not (Fig. 4d). Also, treatment of human cells with HU increased the mono-ubiquitination of p50$^{wt}$, a finding not seen with p50$^{2KR}$ (Fig. 4e). Altogether, these findings indicate that p50 K354 and K356 are mono-ubiquitinated by BARD1 in response to RS and ATR activation.

**ATR and BARD1 modify p50 stability and chromatin recruitment.** Mono-ubiquitination regulates proteolytic and non-proteolytic processes. We examined whether interaction with BARD1 altered p50 protein stability. Wild type and mutant p50

constructs were expressed together with BARD1 in the presence of the protein synthesis inhibitor, cycloheximide (CHX). While BARD1 overexpression increased the half-life (t$_{1/2}$) of p50$^{wt}$, it did not alter the t$_{1/2}$ of p50 constructs that either did not interact with it (p50$^{S337A}$ and p50$^{DE2A}$) or were not mono-ubiquitinated (p50$^{2KR}$) (Fig. 5a and Supplementary Fig. 4a). Consistent with a stabilizing role, knockdown of *BARD1* resulted in decreased endogenous p50 protein (Fig. 5b). The finding that mutation of the ubiquitination sites (2KR) destabilized p50 suggested that mono-ubiquitination itself was important. To directly examine this, we generated p50 fusion constructs expressing a C-terminal ubiquitin. Addition of a single ubiquitin moiety substantially increased the t$_{1/2}$ of mutant p50, a finding particularly striking for p50-DE2A-Ub and p50-2KR-Ub where the t$_{1/2}$ was increased by over 10-fold relative to their respective parental constructs, p50$^{DE2A}$ and p50$^{2KR}$ (Fig. 5c and Supplementary Fig. 4b compared to 4a). These results indicate that mono-ubiquitination of p50 stabilizes nuclear p50 protein.

Mono-ubiquitination is also important in nuclear to cytoplasmic shuttling[33], and has been linked to NF-κB activation by DNA DSBs[34]. We examined whether p50 mono-ubiquitination altered its subcellular localization. Over-expressed p50 distributed to the nucleus and cytoplasm (Fig. 5d), a finding previously reported[11]. However, mutation of either the BARD1-binding motif or the ubiquitination sites resulted in substantially decreased cytoplasmic p50 relative to nuclear (Fig. 5d). Rescue of p50$^{2KR}$ by attachment of a single ubiquitin specifically increased cytoplasmic p50 (Fig. 5e). To further study this, we isolated nuclear and cytoplasmic fractions and examined ubiquitinated p50. Consistent with the distribution of p50 mutants, mono-ubiquitinated p50 protein was found in the cytoplasmic compartment (Fig. 5f). Despite cytoplasmic localization

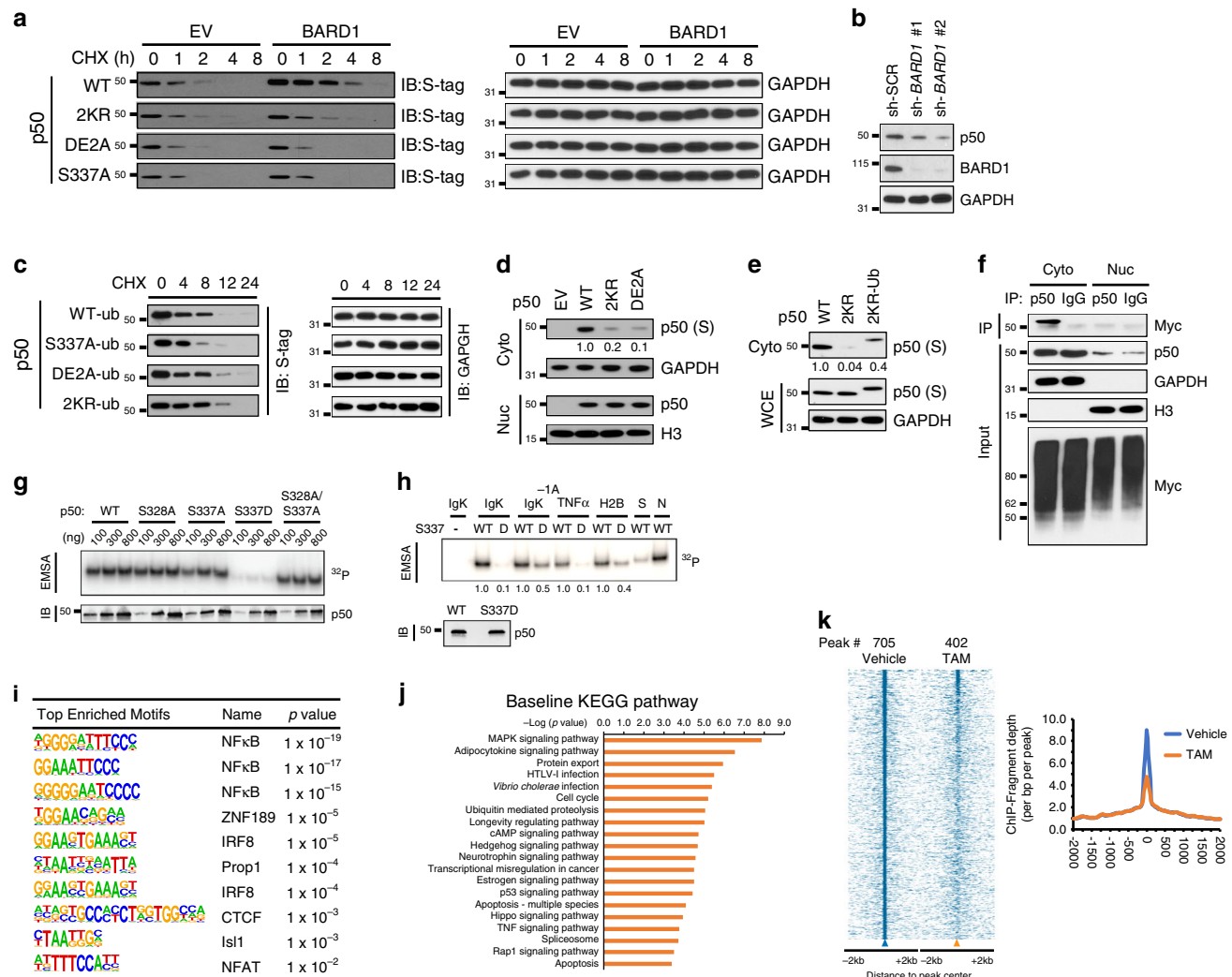

**Fig. 5 Mono-ubiquitination stabilizes p50 and decreases chromatin recruitment. a** *Bard1*-null cells transfected with BARD1 or empty vector (EV) and the indicated S-tag p50 construct. IB performed with anti-S antibody or anti-GAPDH at the indicated time after cycloheximide (CHX) addition. See Supplementary Fig. 4a for quantification. **b** 293T cells infected with sh-RNA targeting *BARD1*. IB performed with indicated antibody. **c** *Bard1*-null cells transfected with BARD1 and the indicated S-tag p50 construct fused with a C-terminal Ubiquitin (0 K and GG/AA mutant Ub). IB performed at the indicated time after CHX addition. See Supplementary Fig. 4b for quantification. **d** 293T cells transfected with the indicated p50 construct were separated into nuclear (nuc) and cytoplasmic (cyto) fractions. IB with anti-S antibody or indicated loading control. **e** 293T cells transfected with S-tag p50$^{wt}$, p50$^{2KR}$, or p50$^{2KR-Ub}$ were fractionated into cytoplasmic (cyto) fraction. Twenty percent of the cell pellet was isolated for analysis of whole-cell extract (WCE). IB was performed with anti-S tag. **f** 293T cells transfected with Myc-Ub were separated into nuc and cyto fractions. IP in each fraction performed with anti-p50 antibody or anti-IgG and IB with anti-Myc. **g** Electrophoretic mobility shift assay (EMSA, upper) or immunoblot (IB, lower) of increasing amounts of bacterially expressed p50$^{wt}$, or p50 mutant, protein using the IgK probe. The same membrane was analyzed by autoradiography ($^{32}$P) and IB. **h** EMSA (upper) or IB (lower) of p50$^{wt}$ (WT) and p50$^{S337D}$ (D) protein using the κB-site probe from the indicated gene (tumor necrosis factor alpha, TNFα; histone 2B, H2B). IB of input proteins demonstrates equal loading. IgK$^{-1A}$ is the IgK with adenine at the −1 position. S and N, specific and non-specific competitor, respectively. **i** DNA motifs most significantly enriched for p50 on ChIP-Seq analysis of 293T cells expressing TopBP1$^{ER}$ treated with vehicle. **j** Kyoto Encyclopedia of Genes and Genomes (KEGG) pathway analysis of p50 ChIP-seq peaks at baseline ranked by −log10 (*P*-value). **k** ChIP-seq analysis of p50-binding peaks. Heatmap (left) and histogram (right) for average peak intensity was plotted for ChIP-Seq data from vehicle and tamoxifen (TAM) treated samples. All blots are representative of at least two biologically independent experiments. Analysis of fold-change normalized to control lane shown below IB where indicated.

of mono-ubiquitinated p50, BARD1 is primarily nuclear and p50 is phosphorylated by CHK1 and interacts with BARD1 in the nucleus. We therefore examined the kinetics of p50 ubiquitination using cells expressing TopBP1$^{ER}$ treated with TAM. In the nucleus, an increase in p50 mono-ubiquitination was seen at 1 h, whereas the kinetics of the appearance of the mono-ubiquitin band was slower in the cytoplasm (Supplementary Fig. 4c). These findings support the hypothesis that mono-ubiquitinated p50 preferentially distributes to the cytoplasm.

Ultimately, p50 mediates its effects by modulating gene expression and, as p50 lacks a transactivation domain, changes in its chromatin recruitment are critical in regulating the downstream response. We therefore examined the effect of p50 mutation on DNA binding. Whereas p50$^{S337A}$ had decreased DNA binding compared to p50$^{wt}$, mutation to S337D completely abrogated binding (Fig. 5g), a finding previously reported[35]. Examination of other κB-sequences demonstrated that while there was generally decreased binding of p50$^{S337D}$, some

differential binding was seen with κB-sites bearing different nucleotide sequences (Fig. 5h). In addition, p50$^{S337A}$ was previously reported to form NF-κB homo- and hetero-dimers to a similar extent as the WT protein[5]. Also, whereas p50$^{DE2A}$ had reduced DNA binding compared to p50$^{wt}$, p50$^{2KR}$ bound κB probes similarly to WT (Supplementary Fig. 4d).

Although p50$^{S337D}$ represents p50 in the phosphorylated state, to examine the in vivo changes in p50 binding induced by ATR on a genome-wide scale, we performed ChIP-seq analysis of endogenous p50 in cells expressing TopBP1$^{ER}$. At baseline, p50 was bound to 705 sites that included regions with κB motifs and other areas (Fig. 5i, Supplementary Fig. 4e and Supplementary Data 1). Pathway analysis demonstrated that the genes linked to these p50 peaks regulate, among other pathways, MAPK and adipocytokine signaling, protein export and the cell cycle (Fig. 5j). Treatment with TAM resulted in decreased p50 enrichment at over 40 % of these sites (Fig. 5k), a finding consistent with the lower DNA binding of p50$^{S337D}$ relative to p50$^{wt}$ on gel shift. Together, these data indicate that the ATR-induced interaction of p50 with BARD1 promotes p50 stabilization and nuclear export and that activation of ATR results in a general decrease in p50 chromatin enrichment.

**p50 interacts with BARD1 to regulate S phase progression.** ATR is not only activated exogenously, but also during S-phase of the cell cycle[36,37]. We examined whether S337 is phosphorylated at this time. Cells were synchronized at G1 by double-thymidine block and then released into media for increasing lengths of time as previously described[13]. Consistent with its role in the G1/S transition[38], Cyclin E increased in early S phase, and in both mouse and human cells p50 S337 phosphorylation was also increased in S phase, with the highest level in early S phase, during the G1/S transition (Fig. 6a and Supplementary Fig. 5a). Notably, this phosphorylation was CHK1-dependent as knock-down of CHK1 blocked S337 phosphorylation (Supplementary Fig. 5b). There was also an increase in the association of p50 and BARD1 beginning in early S phase (Fig. 6b and Supplementary Fig. 5c). This was accompanied in later S phase by p50 mono-ubiquitination (Fig. 6c). To study the role of specific p50 residues in these findings, we expressed WT and mutant p50 and examined cells following release from synchronization. In S phase, BARD1 only interacted with p50$^{wt}$ or p50$^{2KR}$ but not with either p50$^{S337A}$ or p50$^{DE2A}$ (Fig. 6d). Also, only p50$^{wt}$ but none of the p50 mutants, including p50$^{2KR}$, were mono-ubiquitinated in S phase (Fig. 6d). These results indicate that p50 interacts with BARD1 in early S phase via its BRCT interaction motif, and that the association results in p50 mono-ubiquitination at this time.

Mono-ubiquitination of p50 during S phase suggested that this post-translational modification (PTM) might regulate S phase progression. To study this, we examined incorporation of the thymidine analog, 5-ethynyl-2'-deoxyuridine (EdU). Using immortal Nfkb1$^{-/-}$ MEFs, we found that whereas expression of p50$^{wt}$ decreased incorporation of EdU compared to control, p50$^{2KR}$ did not (Fig. 6e). Given this finding, we wanted to ensure that p50$^{2KR}$ was a functional mutant in that it bound DNA. We therefore performed ChIP-seq analysis on cells expressing either p50$^{wt}$ or p50$^{2KR}$ and found that although chromatin enrichment of p50$^{2KR}$ was different than that of p50$^{wt}$, p50$^{2KR}$ was nevertheless able to bind DNA (Supplementary Fig. 5d and Supplementary Data 2). We also examined cellular proliferation and found that while p50$^{wt}$ decreased the growth rate of Nfkb1$^{-/-}$ MEFs, p50$^{2KR}$ and other mutants did not (Fig. 6f and Supplementary Fig. 5e). In addition, as it was previously shown that p50 restricted tumor growth in mice[9], we examined the role of the p50/BARD1 interaction in this response by expressing

p50$^{wt}$ or p50$^{DE2A}$ in transformed Nfkb1$^{-/-}$ MEFs and injecting the cells into nude mice. p50$^{wt}$ reduced tumor size compared to empty vector, but p50$^{DE2A}$ did not (Fig. 6g), indicating that the propensity of p50 to restrict tumor growth was mediated by its ability to bind BARD1. Finally, we examined whether adding an ubiquitin moiety to mutant p50 altered cell cycle progression. Covalent addition of a single ubiquitin to p50$^{2KR}$ significantly reduced both the percentage of cells in S-phase and overall cellular proliferation relative to parental p50$^{2KR}$ and almost to the level seen with p50$^{wt}$ (Fig. 6h and Supplementary Fig. 5f). Together, these findings indicate that p50 interaction with BARD1 and mono-ubiquitination act to slow S phase progression.

To identify downstream factors that mediate the effect of p50 on the cell cycle, we interrogated the p50 ChIP-Seq peaks in cells expressing TopBP1$^{ER}$ (Fig. 5k) with factors known to be involved in the cell cycle. Given the cyclical change in p50 PTM during the cell cycle, we specifically looked at factors whose expression levels are periodically modulated during the cell cycle (CYCLEBASE 3.0, Supplementary Data 3)[39]. Of these periodic genes, twelve were linked to p50 peaks bound only at baseline and six were linked to peaks bound only following TAM stimulation (Fig. 6i and Supplementary Data 3). Although these genes represent cell cycle factors regulated by p50, to more specifically identify genes modulated by p50 during S phase transition, we examined changes in p50 chromatin recruitment in G1 and S phase. Whereas p50 was bound to 1212 peaks in G1, this was reduced to 611 peaks in S phase (Fig. 6j, Supplementary Fig. 6a, b and Supplementary Data 4), a finding consistent with the decreased p50 recruitment seen following ATR activation (Fig. 5k). We then compared these p50 peaks to the cell cycle genes regulated in response to ATR in TopBP1$^{ER}$ cells. Two genes, CCNE1 and DNAJB6, were linked to p50 peaks bound in G1 but not S phase and in TopBP1$^{ER}$ cells at baseline but not following TAM stimulation (Supplementary Fig. 6c and Supplementary Data 3). In other words, during progression from G1 to S phase, or following ATR activation, p50 binding to the promoters of these two genes was decreased. Of these two, CCNE1 was conspicuous because of its critical role in regulating G1/S transition. To validate the changes in p50 binding during the cell cycle, we performed quantitative ChIP-PCR and found that p50 recruitment to the CCNE1 promoter was significantly decreased in S phase relative to G1 (Fig. 6k), a finding recapitulated at the DNAJB6 but not the KPNA2 promoter (Supplementary Fig. 6d). Finally, we examined whether p50 and its PTM altered Cyclin E level. Whereas p50$^{wt}$ decreased Cyclin E relative to control, p50$^{2KR}$ had much less effect (Fig. 6l). Moreover, expression of the p50 motif-mutant, p50$^{DE2A}$, also failed to repress cyclin E in tumors (Supplementary Fig. 6e). Taken together, these results indicate that during the cell cycle, PTM of p50 at the G1/S transition reduces its binding to the CCNE1 promoter facilitating Cyclin E expression and S phase progression. Consistent with this, mutation of p50 increased both Cyclin E and cellular proliferation.

**p50 PTM is required for genome maintenance and p50 protein correlates with BARD1 in human cancer.** Increased cellular proliferation and Cyclin E deregulation induce RS and genome instability that promote cancer. We examined whether mutation of p50 mono-ubiquitination sites affect genome stability. Primary Nfkb1$^{-/-}$ MEFs were freshly harvested and infected with vectors expressing either p50$^{wt}$ or p50$^{2KR}$. These cells were then treated with low-dose aphidicolin (APH) to induce RS or serially passaged, and chromosomal breaks and gaps scored. Expression of p50$^{2KR}$ led to significantly higher numbers of breaks and gaps

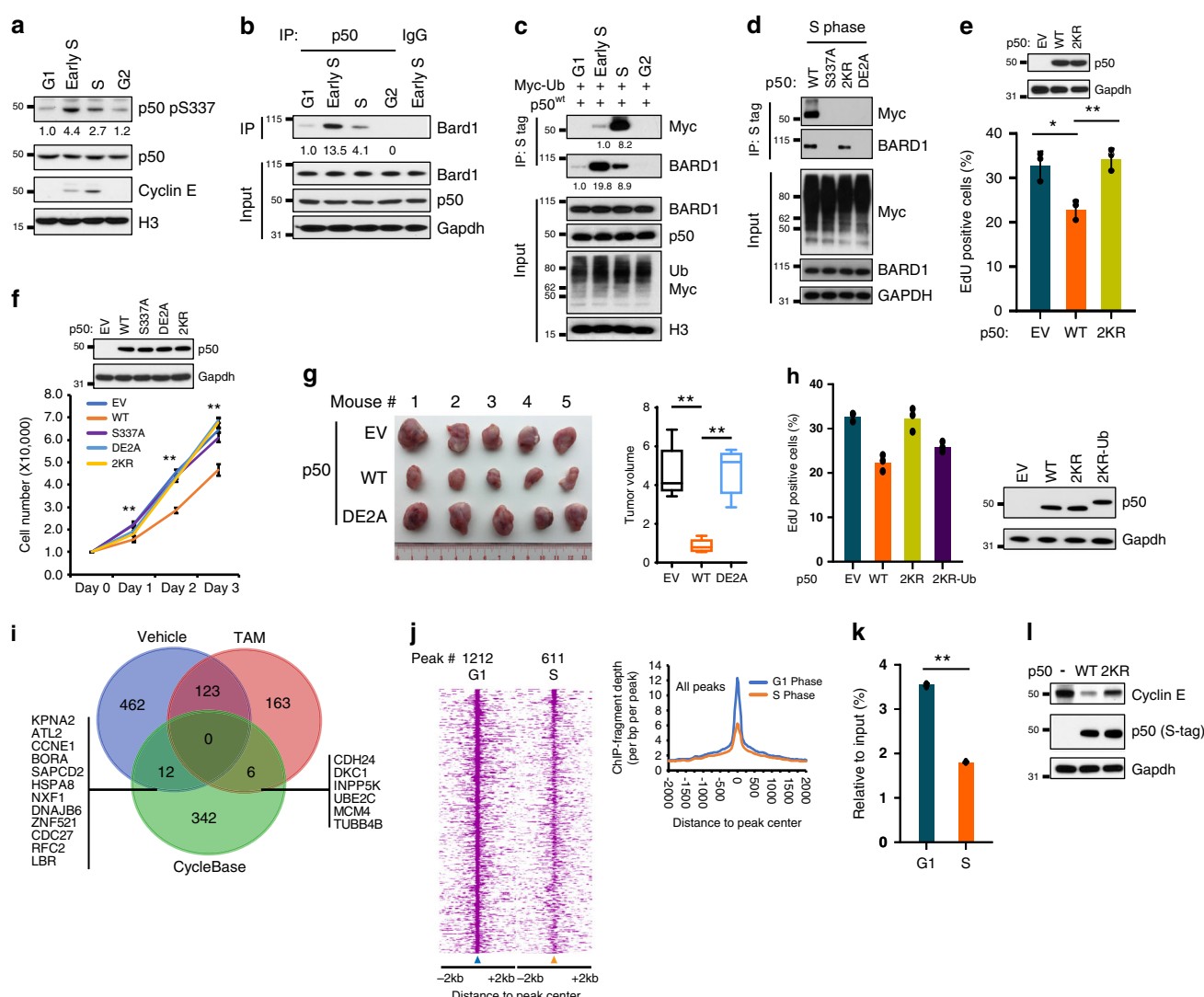

**Fig. 6 p50 mono-ubiquitination regulates cell cycle progression. a** Synchronized primary WT MEFs were harvested at the indicated cell cycle stage and nuclear fractions isolated. IB using anti-p50-pS337, anti-p50, and anti-Cyclin E. **b** Co-IP in primary WT MEFs synchronized and released as in **a**. IP with anti-p50 or IgG and IB with anti-Bard1 or indicated antibody. **c** 293T cells transfected with Myc-Ub and S-tag p50$^{wt}$ were harvested at the indicated cell cycle stage and nuclear extract used for IP with S-agarose followed by IB with anti-Myc or anti-BARD1 antibody. **d** 293T cells transfected with Myc-Ub and p50 constructs were harvested at S phase. IP of nuclear fractions with S-agarose and IB with anti-Myc or anti-BARD1. **e** Immortal $Nfkb1^{-/-}$ MEFs stably expressing empty vector (EV), p50$^{wt}$ (WT), or p50$^{2KR}$ (2KR) were pulsed with 5-ethynyl-2 deoxyuridine (EdU) and positive staining quantified. Data show mean value from $n$ = three biologically independent experiments, ± SEM. *$P$ = 0.012, **$P$ = 0.004, two-sided Student's $t$-test. Inset: IB with anti-p50. **f** Immortal $Nfkb1^{-/-}$ MEFs stably expressing EV or the indicated p50 mutant were plated and cell number counted every 24 h. Data show mean number from three biologically independent experiments, ± SEM. **$P$ < 0.001, EV vs. WT, two-sided Student's $t$-test. **g** Immortal $Nfkb1^{-/-}$ MEFs stably expressing the indicated construct were injected into flanks of nude mice and tumors harvested at 2 weeks. Representative experiment (left panel, $n$ = 5 animals). Box plot: center-line is median, box limits represent upper and lower quartiles and whiskers are minimum and maximum. **$P$ < 0.001, two-sided Student's $t$-test. **h** Immortal $Nfkb1^{-/-}$ MEFs stably expressing EV, p50$^{wt}$, p50$^{2KR}$, or p50$^{2KR-Ub}$ (2KR-Ub) were pulsed with EdU and positive staining quantified. Data show mean value from two biologically independent experiments. **i** Venn diagram illustrating overlap between genes associated with p50-binding peaks in ChIP-Seq analysis from Fig. 5 and periodically regulated genes from Cyclebase 3.0. **j** ChIP-seq analysis of p50-binding peaks. Heatmap (left) and histogram (right) for average peak intensity was plotted for ChIP-Seq data from cells synchronized at G1 and S phase. **k** ChIP-qPCR of p50 binding to the *CCNE1* promoter in 293T cells at the indicated cell cycle phase. Data show mean enrichment of p50 relative to input and IgG control, ± SEM. from $n$ = 3 biologically independent experiments.**$P$ < 0.001, two-sided Student's $t$-test. **l** IB using $Nfkb1^{-/-}$ MEFs expressing EV, S-p50$^{wt}$ or S-p50$^{2KR}$ using anti-Cyclin E and anti-S tag. All blots are representative of at least two biologically independent experiments. Analysis of fold-change normalized to control lane shown below IB where indicated.

compared to p50$^{wt}$ both at baseline and following APH treatment (Fig. 7a). Moreover, following serial passage, primary $Nfkb1^{-/-}$ MEFs expressing p50$^{2KR}$ accumulated more spontaneous breaks than MEFs expressing p50$^{wt}$ (Supplementary Fig. 7a). These findings suggest that p50 mono-ubiquitination is required for maintaining genome stability.

Mono-ubiquitination of p50 required interaction with the BARD1 BRCT domains. Many cancer-associated *BARD1* mutations localize to its BRCT domains. We examined whether cancer-associated BRCT mutations affect the interaction of BARD1 with p50. Although p50 bound BARD1 mutated at residues outside the BRCT domains (C557S and Q564H), p50 did

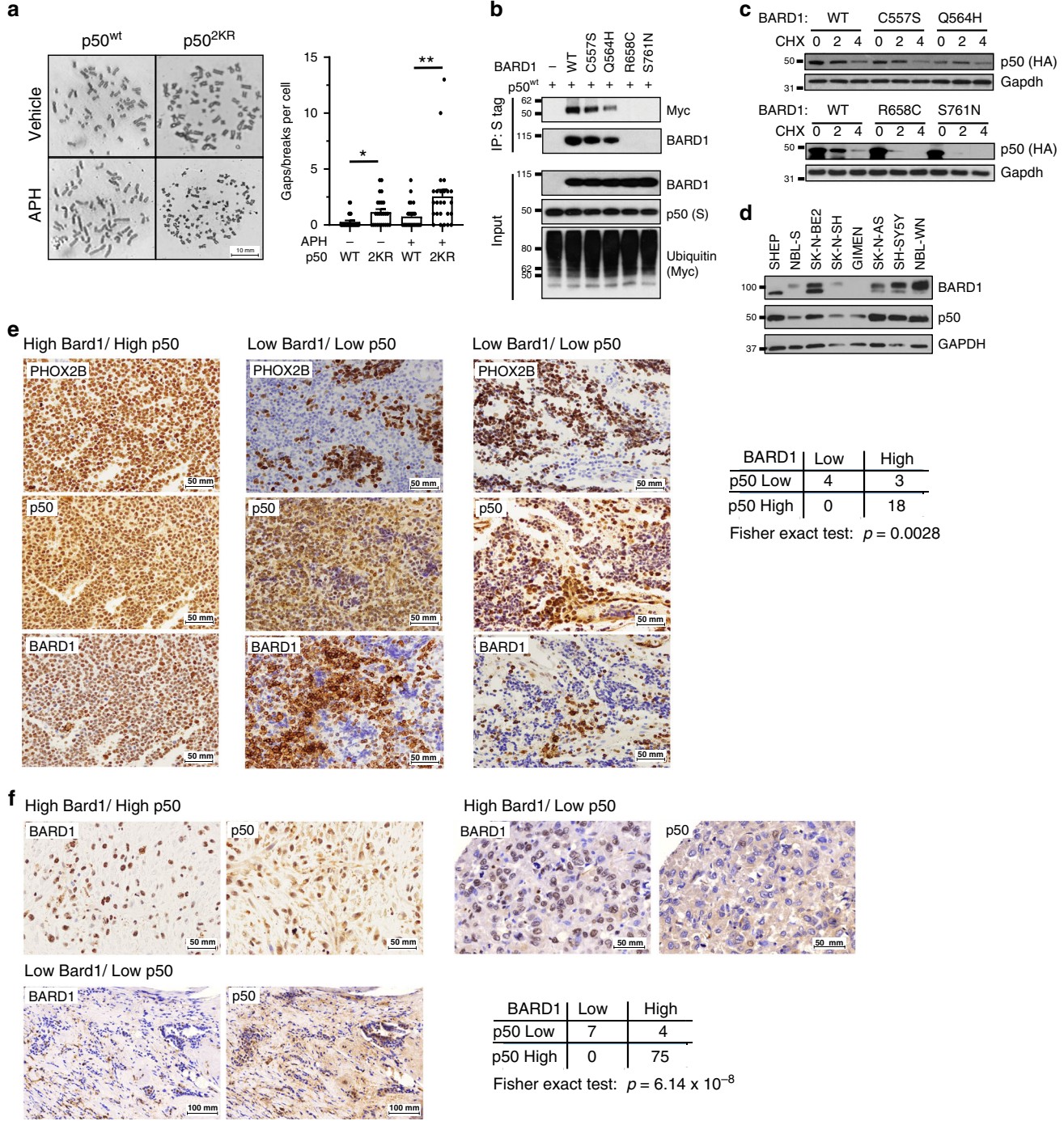

**Fig. 7 BARD1 and p50 protein in human cancer. a** Analysis of breaks and gaps in primary *Nfkb1*⁻/⁻ MEFs expressing either p50^wt or p50^2KR. Cells were treated with vehicle or aphidicolin (APH, 0.1 μM 24 h). Data show gaps or breaks from 25 random metaphase cells, ± SEM. *$P = 0.003$, **$P = 0.007$, two-sided unpaired Student's *t*-test. Representative metaphase spreads shown (left). **b** Co-IP in *Bard1*-null mouse cells transfected with Myc-Ub, S-tag p50^wt and either empty vector or BARD1 wild type or mutant as indicated. IP with S-agarose was performed on nuclear fractions and analyzed by IB with anti-Myc or anti-BARD1 antibody. **c** *Bard1*-null mouse cells transfected with HA-p50^wt and the indicated BARD1 construct. Cells were treated with cycloheximide (CHX) and harvested at the indicated time. IB with anti-HA antibody. **d** Neuroblastoma cell lines analyzed by IB with anti-BARD1 (A300-263A) or anti-p50 antibody. **e** Representative IHC staining of p50 and FL BARD1 in neuroblastoma patient samples. Only regions with paired-like homeobox 2b (PHOX2B)-positive cells considered in p50/BARD1 scoring. Left—high BARD1/high p50, all cells are PHOX2B positive (brown) and have high nuclear p50 and BARD1. Center and right—low BARD1/low p50 from two independent patients, regions that are positive for PHOX2B (brown) have low (blue) nuclear p50 and low BARD1. Table shows numbers of tumors in each category analyzed by two-sided Fisher's exact test. **f** Representative IHC staining of p50 and FL BARD1 in human breast cancer TMAs. Only nuclear p50 and BARD1 examined. Left upper—high BARD1/high p50; right upper—high BARD1/low p50, left lower—low BARD1/low p50. Table (right lower) shows numbers of tumors in each category analyzed by two-sided Fisher's exact test. All blots are representative of at least two biologically independent experiments.

not bind BARD1 with cancer-associated mutations within the BRCT domains (R658C and S761N, Fig. 7b). Consistent with this, p50 was not mono-ubiquitinated in the presence of the BRCT mutants, R658C and S761N (Fig. 7b). Moreover, while the BARD1 mutants, C557S and Q564H, stabilized p50 protein to a similar extent as WT BARD1, R658C and S761N did not stabilize p50 (Fig. 7c and Supplementary Fig. 7b). These data indicate that mutations of the *BARD1* BRCT domains found in human cancer block the ability of BARD1 to bind and stabilize p50, an observation similar to that seen in the presence of reduced BARD1 protein.

The above findings suggest that the ability of BARD1 to stabilize p50 might be important in cancer, raising the question of whether there is an inherent correlation between these proteins in human tumors. Neuroblastoma is one cancer closely associated with *BARD1*[40,41]. We obtained a series of neuroblastoma cell lines and examined BARD1 and p50 protein in these cells. Consistent with the experimental manipulation data, cells with lower amounts of BARD1 generally had less p50 protein (Fig. 7d and Supplementary Fig. 7c). Moreover, expression of BARD1 in GIMEN cells that have particularly low BARD1 led to increased endogenous p50 protein (Supplementary Fig. 7d). Given this link between endogenous p50 and BARD1, we next looked at these two proteins in clinical tumor specimens using immunohisto-chemistry (IHC). Importantly, BARD1 has several isoforms that lack specific motifs such as the RING domain. As many of these isoforms cannot stabilize p50, it was important to specifically look at full length (FL) BARD1. To this end, we used a previously described mouse monoclonal antibody raised against the N-terminal of FL BARD1[42]. We verified that this antibody only recognized FL BARD1 and not other isoforms by immunoblot in neuroblastoma cell lines (Supplementary Fig. 7e). Subsequently, we obtained 25 verified neuroblastoma specimens from the University of Chicago pathology archive and examined nuclear BARD1 and p50 in contiguous sections. By looking specifically in the nucleus, we minimized cross-reactivity of anti-p50 with its precursor p105, given that p105 is cytoplasmic. IHC grading was performed in a blinded fashion based on a four-tier system that was converted into a binary scale as previously described[43]. Importantly, to validate the presence of neuroblastoma cells within the sections, the neuroblastoma-specific marker, paired-like homeobox 2B (PHOX2B)[44], was examined, and only staining in the PHOX2B-positive cells was considered in the analyses. In these specimens, a significant positive correlation between nuclear FL BARD1 and p50 was noted specifically in PHOX2B-positive cells ($P = 0.0028$, Fisher's exact test) (Fig. 7e). Notably, while a few specimens with high BARD1 staining had low p50 (Supplementary Fig. 7f), all tumors with low BARD1 staining had low nuclear p50.

Breast cancer has also been linked to BARD1. We examined the correlation between nuclear FL BARD1 and p50 in a series of breast cancer specimens. Three tissue microarrays (TMAs), containing tumor and normal breast tissue from ninety-one patients, were constructed as previously described[45]. In breast cancer specimens, IHC staining was graded based on both staining proportion and intensity and the overall score dichotomized as either high or low as previously described[46]. In this cancer, a highly significant positive correlation was seen between the amount of nuclear FL BARD1 and nuclear p50 protein ($P = 6.14 \times 10^{-8}$, Fisher's exact test). Notably, whereas the majority of patients had high nuclear p50 and BARD1, all the patients who had low nuclear BARD1 also had low nuclear p50 (Fig. 7f). Altogether, these findings support the experimental manipulation studies and demonstrate that in clinical cancer specimens, the presence of low nuclear BARD1 is strongly correlated with low nuclear p50 protein. Finally, we examined whether there was a correlation between p50 or BARD1 staining and patient survival. Remarkably, when outcome was dichotomized to alive or dead, low nuclear p50 staining was significantly associated with death ($P = 0.048$, Fisher's exact test, Supplementary Fig. 7g). Unfortunately, insufficient numbers of patients with low BARD1 had survival data for meaningful analysis.

## Discussion

Mounting evidence suggests that the mature product of *NFKB1*, p50, promotes genome stability and tumor suppression. To elucidate the mechanism by which p50 acts in this manner, we used affinity purification coupled with MS/MS and identified BARD1 as a p50-interacting factor. p50 directly associated with the BARD1 BRCT domains independent of BRCA1. While prior studies have demonstrated an association between BRCA1 and NF-κB, the exact mechanism of this interaction is unclear. In one study, BRCA1 interacted with p65[47], whereas in another report BRCA1 facilitated p50 promoter recruitment independent of p65[48].

A potential BARD1 BRCT phospho–protein-binding motif was previously described[27], although not confirmed[30]. We identified a conserved sequence in p50 based on S337 that loosely conformed to this motif. Mutation of either S337 or the motif itself (DE2A) blocked binding of p50 to BARD1. S337 is a highly conserved amino acid that is homologous to the critical p65 residue, S276[49]. Although protein kinase A (PKA) was previously reported to phosphorylate S337 in vitro[35], no functional role for this PTM had previously been demonstrated. We found that phosphorylation of S337 promoted interaction of p50 with BARD1, leading to p50 mono-ubiquitination at two C-terminal lysines. While BARD1 overexpression also increased p50 poly-ubiquitination, loss of the interaction with BARD1 specifically blocked mono-ubiquitin addition without altering poly-ubiquitination. In addition, although p50 interaction with BARD1 was independent of BRCA1, mono-ubiquitination required BRCA1, a finding consistent with the requirement of the entire BARD1/BRCA1 complex for efficient enzymatic activity[14]. From a functional perspective, mono-ubiquitination stabilized p50 protein and promoted its nuclear export.

Phosphorylation of S337 was induced by CHK1 in response to ATR activation. We previously reported that this pathway also results in phosphorylation of p50 at S328[12,13]. Although we found that mutation of S328 did not affect S337 phosphorylation, the exact relationship between p50 S328 and S337 phosphorylation remains unclear. The initial MS/MS data suggested that S328 mutation blocks the interaction of p50 with BARD1. Given the requirement of interaction with BARD1 for p50 mono-ubiquitination, we speculate that loss of S328 phosphorylation also blocks p50 mono-ubiquitination.

ATR and CHK1 are activated during S phase of the cell cycle. Consistent with this, p50 was phosphorylated in early S phase. Moreover, p50 interacted with BARD1 and was mono-ubiquitinated at this time. Genome-wide studies demonstrated that the chromatin recruitment of p50 was substantially decreased in S phase relative to G1, and the *CCNE1* promoter was identified as a region bound by p50 in G1 and released in S phase. Consistent with the general inhibitory effect of p50 dimers, over-expression of wild type p50 decreased Cyclin E, a finding not seen with mutation of the p50 mono-ubiquitination sites. Notably, the *CCNE* promoter was previously reported to contain a κB-site and to be negatively regulated by NF-κB[50]. Functionally, loss of p50 PTM resulted in increased S phase progression. Altogether these findings suggest a model in which p50 is bound to promoters in G1 acting to attenuate their expression. Upon activation of ATR in early S phase, PTM of p50 results in a decrease in its DNA

binding facilitating the increase in Cyclin E that promotes cell cycle progression. In cells that have reduced p50, or reduced p50 PTM, steady state Cyclin E level and cell proliferation were increased. In addition, consistent with the observation that S337 phosphorylation leads to a decrease in p50 DNA binding, we found that the p50 phosphomimetic, p50[S337D], also has drastically reduced DNA binding compared to p50[wt]. Interestingly, enhanced proliferation and reduced time in G1 were previously described in cells deleted of Nfkb1[51]. Ultimately, overexpression or deregulation of Cyclin E is an oncogenic stimulus that promotes RS and genome instability[52,53]. Consistent with this, we found that mutation of the p50 mono-ubiquitination sites resulted in increased chromosomal aberrations indicating that loss of this pathway leads to chromosomal damage and genome instability.

BARD1 is also important for genome stability and several cancer-associated BARD1 missense mutations localize to its BRCT domains. While these domains have been reported to associate with poly(ADP-ribose) (PAR) moieties and with HP1γ in the setting of DNA damage[54,55], no phospho-proteins have yet been shown to interact with them. The current work identifies p50 as one such factor. Interaction with BARD1 was seen even in the absence of exogenous damage during the unperturbed cell cycle. This association stabilized p50 protein such that either decreased FL BARD1 or BARD1 mutation resulted in decreased nuclear p50 protein. Consistent with this, we found that the cancer-associated BARD1 BRCT mutants, R658C and S761N, failed to interact with p50 or stabilize it. Notably, these mutants, like p50 itself, do not modulate HDR[13,26,56]. A potential mechanism by which S761N might reduce phospho–protein binding was previously proposed[30]. While these findings support a role for the phospho-p50/BARD1-BRCT interaction in maintaining genome stability, whether phospho–protein binding is critical for tumor suppression by BARD1 is unclear. In this regard, it was recently reported that mice with mutations in the Bard1 phosphate-binding pocket were not tumor prone although they did demonstrate evidence of chromosome instability[57]. The Bard1 mutations in these mice correspond to critical phospho–protein binding residues in Brca1 and abrogated Bard1 binding to PAR. Although their interaction with a phospho-peptide has not previously been reported, we found that p50 did not interact with BARD1-S575A, one of the residues mutated in the knock-in mice (although in the mouse the mutation was Ser to Phe)[57]. Given the phospho-dependent interaction of p50 with BARD1, it would be interesting not only to examine p50 in these mice, but also to evaluate whether they have altered susceptibility to carcinogen-induced tumor formation, as was previously seen with Nfkb1[-/-] mice[4].

The stabilizing effect of BARD1 on p50 raised the question of whether there was an inherent correlation between these factors in human cancer. We examined the relative abundance of these proteins in tumors linked to BARD1. A strong positive correlation between nuclear BARD1 and p50 protein was seen in both neuroblastoma and breast cancer specimens. Specifically, tumors with low nuclear BARD1 also had low nuclear p50. Interestingly, we found that about 10% of our breast cancer specimens had low nuclear BARD1. Although BARD1 mutation is generally found in <1% of breast cancers, BRCA1 mutation is seen in up to 10% of tumors[58]. Given that BRCA1 stabilizes BARD1 protein[14], the high percentage of these tumors with low BARD1 is likely related to BRCA1 loss. Interestingly, in our limited cohort of breast cancer specimens, we noted a significant correlation between low nuclear p50 staining and earlier death, a finding that supports the deleterious effects of reduced nuclear p50.

Although NF-κB is best known as a stimulus-induced transcription factor, even at baseline there is a significant amount of nuclear, DNA-bound p50. This nuclear p50 does not require prior stimulation-induced translocation to modulate gene expression and, therefore, is ideally positioned to rapidly regulate routine cellular responses. In this regard, we found that p50 PTM during the cell cycle altered its DNA binding facilitating the increase in Cyclin E required for S phase progression. Loss, or decrease, in nuclear p50 changed this homeostasis resulting in deregulation of the cell cycle, induction of RS and increased chromosomal aberrations. Such a physiological role for p50 suggests that its loss is not only relevant to cancer but also to other medical conditions. This observation is underlined by the premature aging and increase in chronic disease seen with loss of this subunit in both mice and humans[59–62].

## Methods

**Cell culture**. HEK293T, HeLa, and MCF-7 cell lines were obtained from ATCC, and were routinely passaged in Dulbecco's modified Eagle's medium (DMEM). Human neuroblastoma cell lines (SK-N-AS, NBL-S, SHEP, SK-N-DZ, SK-N-BE2, GIMEN, IMR5, NGP, SK-N-SH, SH-SY5Y, NBL-WN) were a gift from Dr. S Cohn (The University of Chicago) and cultured in RPMI 1640 media. The BRCA1 mutant cell line HCC1937 was a gift from Dr. P Connell (The University of Chicago) and was passaged in DMEM high-glucose media. Bard1-null mouse mammary carcinoma cell line (10-05) was a gift from Dr. R Baer (Columbia University Medical Center) and was passaged in DMEM high-glucose media. Immortal Nfkb1[-/-] MEFs have been described[12] and primary WT and Nfkb1[-/-] MEFs were freshly harvested at embryonic day 13.5. All media were supplemented with 10% fetal bovine serum (FBS) (Atlanta Biologicals) and 1% penicillin/strep-tomycin/L-glutamine (ThermoFisher Scientific). Cells were routinely screened for potential Mycoplasma contaminations.

**Immunoblots and antibodies**. Cells were harvested and lysed by RIPA buffer (ThermoFisher Scientific) supplemented with Halt Protease and Phosphatase Inhibitor (ThermoFisher Scientific) and total protein quantified by Bradford protein quantification assay (Bio-Rad). Equal amounts of protein sample were mixed with 4x sample buffer (Bio-Rad) and boiled for 5 min. The sample was subjected to SDS–PAGE and transferred to polyvinylidene fluoride membrane (Immobilon-P, Millipore-Sigma). After blocking with 5% non-fat dry milk (5%BSA for phosphorylation antibody) for 1 h at room temperature (RT), primary antibody was added at 4 °C and left overnight. After washing, the membrane was incubated with horseradish peroxidase-conjugated secondary antibodies for 1 h at RT. Protein bands were visualized by SuperSignal West Femto Peroxide Solution (Thermo-Fisher Scientific). All blots are representative of at least two independent experiments.

The following primary antibodies were used: anti-FLAG M2 antibody (1:20,000, Sigma-Aldrich, F1804); anti-FLAG rabbit monoclonal antibody (1:10,000, Cell Signaling, #14793), Anti-HA tag antibody ChIP Grade (1:20,000, Abcam, Ab9110); anti-S-Tag antibody (1:10,000, BioLegend, 688102); anti-Myc tag antibody (1:5,000, Santa Cruz, sc-40); anti-BARD1 mouse monoclonal antibody (1:500, Santa Cruz, sc-74559); anti-BARD1 rabbit polyclonal antibody (1:2,000, Bethyl Laboratories, A300-263A); anti-BARD1 rabbit polyclonal antibody (1:1,000, Santa Cruz, sc-11438); anti- NFκB1/p50 mouse monoclonal antibody (1:500, Santa Cruz, sc-8414X); anti-BCL-3 antibody (1:500, Santa Cruz, sc-185); anti- cyclin E antibody (1:2,000, Santa Cruz, sc-198); anti- BRCA1 antibody (1:500, Santa Cruz, sc-6954); anti- Chk1 antibody (1:2,000, Santa Cruz, sc-8408); anti-GAPDH antibody (1:20,000, Santa Cruz, sc-32233); anti-Histone H3 antibody (1:50,000, Santa Cruz, sc-517576).

The following secondary antibodies were used: donkey anti-Mouse IgG (H + L) Highly Cross-Adsorbed Secondary Antibody, HRP (1:20,000, ThermoFisher Scientific, A16017) and donkey anti-Rabbit IgG (H + L) Highly Cross-Adsorbed Secondary Antibody, HRP (1:20,000, ThermoFisher Scientific, A16035) Where indicated, transfection was performed with TransIT-LT1 Transfection Reagent (Mirus Bio) according to manufacturer's instructions.

**Co-immunoprecipitation**. The indicated cells were harvested using a standard trypsin protocol and lysed on ice with immunoprecipitation (IP) buffer (50 mM Tris-HCl, pH 7.4, 150 mM NaCl, 10 mM NEM, 1% Triton X-100, 1% sodium deoxycholate and 1% protease inhibitor cocktails) for 15 min. Cell lysate was centrifuged for 10 min at 20,800 × g and 4 °C, and the supernatant was incubated with primary antibodies and Dynabeads Protein G (ThermoFisher Scientific) rotating at 4 °C overnight. Where indicated, lysate was treated with vehicle or lambda protein phosphatase (New England BioLabs) prior to IP. The next day, the beads/antibody complex was washed six times with cold IP buffer and subjected to immunoblot analysis.

**Affinity purification and mass spectrometry**. Empty vector, HA-tagged wild type p50 (p50[wt]) or p50[S328A] were expressed in 293T cells. IP was performed as above

and lysate separated on SDS–PAGE gel. Gels were then silver stained with Silver Stain Plus Kit (Bio-Rad) and the indicated band cut out and sent for mass spectrometry (LC-MS/MS) at the Mass Spectrometry and Proteomics Core Facility (Scripps Research Institute).

**Cell fractionation**. Cells were suspended in hypotonic buffer (20 mM Tris-HCl, pH 7.5, 10 mM potassium chloride (KCl), 1.5 mM magnesium chloride (MgCl₂), 1% Triton X-100, and protease inhibitor cocktail) for 10 min. The cytoplasmic fraction was collected by centrifuging the lysate at $1000 \times g$ for 3 min, and further clarified by centrifugation at $20,800 \times g$ for 10 min to remove any potential nuclear contaminations. The nuclear pellet was washed once with hypotonic buffer to remove cytoplasmic debris and the nuclei lysed with radioimmunoprecipitation assay (RIPA) buffer containing protease/inhibitor cocktail.

**Cell cycle synchronization**. For double-thymidine synchronization at the G1/S boundary, cells were cultured in the presence of 2 mM thymidine (Sigma-Aldrich) for 16 h, released for 8 h into medium without thymidine and then blocked again with 2 mM thymidine for another 16 h. Following release into regular media, progression into different phases was confirmed by flow cytometry. Following release, cells entered S phase at 2 h (early S), were in S phase at 4 h and in G2 by 8 h.

**Metaphase spreads**. For analysis of metaphase spreads, freshly harvested primary *Nfkb1⁻ᐟ⁻* MEFs infected with vectors expressing the indicated construct were used. Total breaks, gaps or constrictions on chromosomes were quantified on metaphase spreads stained by Giemsa (Sigma-Aldrich) essentially as previously described[63]. Low-dose (0.1 µM) aphidicolin (Sigma-Aldrich) was added for 24 h where indicated. Samples were scored in a blinded manner by two independent observers and data represent analysis of at least two separate clones. At least 25 separate metaphases were analyzed for each condition.

**EdU incorporation assay**. EdU incorporation was performed with the Click-iT EdU Alexa Fluor 488 Imaging Kit (ThermoFisher Scientific). Immortal *Nfkb1⁻ᐟ⁻* MEFs expressing the indicated construct were pulsed with EdU (10 µM final concentration) for 15 min, fixed in paraformaldehyde and EdU detected according to the manufacturer's instructions.

**Cell proliferation assay**. 293T cells or immortal *Nfkb1⁻ᐟ⁻* MEFs expressing the indicated p50 constructs were seeded in triplicate at a density of $1 \times 10^5$ cells/well in 6-well plates. Cells were then counted under light microscope at 24, 48, and 72 h after seeding.

**Ubiquitin fusion constructs**. To study the function of p50 mono-ubiquitination, we constructed p50-fusion vectors expressing p50^wt or the indicated mutant (p50^2KR, p50^DE2A and p50^S337A) fused to a C-terminal Ubiquitin moiety. The attached Ubiquitin contained G75A/G76A mutations to prevent further covalent binding to other proteins and also had all their lysines mutated (0KUb). These p50 constructs also contained N-terminal S-tag for IB analysis.

**Electrophoretic mobility shift assay (EMSA)**. EMSA was performed using double-stranded oligonucleotide (oligo) probes synthesized (IDT) to contain the decameric κB-site sequence from the promoters of the indicated target genes. Oligos were end labeled with [γ-³²P] adenosine triphosphate (ATP) using T4 polynucleotide kinase. EMSA was performed by incubating oligo probe either with bacterially expressed wild type and mutant p50 protein or with 5 µg of nuclear extract from cells expressing the indicated p50 vector. Absence of nuclear extract, competition with 100-fold molar excess unlabeled NF-κB probe or non-specific oligo served as controls. Complexes were separated by electrophoresis on non-denaturing 5% acrylamide gel and assayed by autoradiography. Each experiment was repeated at least twice and representative images are shown. Sequences of EMSA probes are provided (Supplementary Table 2).

**Chromatin immunoprecipitation assay**. ChIP followed by quantitative real-time PCR (qPCR) was performed at the indicated cell cycle phase using the indicated cells. Cells were released from synchronization and at the indicated cell cycle phase they were fixed with 1% formaldehyde. DNA was sonicated and centrifuged at 20,800 ×g and 4 °C. Supernatant from $1 \times 10^6$ cells was used for each ChIP assay using Dynabeads Protein G or Protein A (ThermoFisher Scientific). Two micrograms of non-immune IgG (sc-2027, Santa Cruz), anti-p50 (sc-7178x, Santa Cruz) antibodies were used per ChIP. Immunoprecipitated DNA was analyzed by real-time qPCR with SYBR Green I. Primer sequences are shown. Non-immunized IgG was used as the negative control for validation of non-specific binding at the various binding sites. The immunoprecipitated DNA was amplified by real-time PCR. All antibodies demonstrated significant enrichment compared to IgG, indicating the high specificity of pulldown by the corresponding antibody. The relative enrichment of the different subunits at each site was expressed as the percentage of

the corresponding input sample (ΔCT). Experiments were performed in triplicate and repeated at least twice.

**ChIP-Seq library construction and data analysis pipeline**. The procedure for ChIP was performed as described above using the indicated antibodies. For pull-down of HA-p50 constructs, ChIP-grade anti-HA antibody (Abcam, ab9110) was used. ChIP-Seq libraries were constructed with NEBNext Ultra II DNA Library Prep Kit for Illumina (NEB). Libraries were then sent to Genomics Core Facility at the University of Chicago for quality control fragment analysis and sequencing with HiSeq4000 platform.

Single-end 50 bp short reads were retrieved from the Genomics Core Facility. Samples were first trimmed with Trimmomatic (v0.38) to remove low-quality reads. Subsequently, reads generated from PCR amplification were filtered with Broad Institute Picard Tools (v 2.18.27). The short reads were mapped to human genome (hg38) with Bowtie2 (v2.3.4.3), and sorted with Samtools (v1.9). Homer pipeline (v4.10) was used for peak calling (findPeaks) and annotation (annotatePeaks.pl). the false discovery rate (FDR) was set as default (0.001). The bedGraph file used for displaying peaks within UCSC Genome Browser was generated with the makeUCSCfile package from Homer. KEGG pathway enrichment analysis was performed with the findGO.pl module in Homer. Heatmaps for binding around peak centers and histograms for average peak intensity were plotted with the annotatePeaks module in Homer suite. The libraries from input DNA were used as controls to enable removal of background random clusters of reads. Robust peaks with peak score ≥10 were used to consolidate peaks to the nearest genes.

**Custom phospho-p50-S337 antibody**. Rabbit polyclonal antibody for detection of p50 S337 phosphorylation was produced by YenZym Antibody, LLC. A peptide containing pS337 (QLRRK-pS-DLETSEPK C- amide) was synthesized and used for immunization. The antibody was then purified with affinity purification. The purified antibody was further affinity-absorbed against the non-phosphorylated peptide to isolate the phosphopeptide-specific antibody from the cross-reactive population.

**Bacteria p50 protein expression**. DNA for human p50 wild type (p50^wt), p50^S328A, p50^S328D, p50^S337A, p50^S337D, p50^S328A/S337A and p50^DE2A was sub-cloned into pET45b + (Novagen) vector fused with an N-terminal 6x polyhistidine (6His) tag. 6His-tagged p50 was then isolated from BL21 *E.coli* using QIAGEN Ni-NTA spin columns. Protein content was quantified by Bradford protein quantification assay (Bio-Rad) and purity confirmed by PAGE and Coomassie staining.

**Glutathione S-transferase (GST) pulldown assay**. DNA for the BRCT domains of human BARD1 was fused with a GST tag and the construct subcloned into the bacterial expression vector, pETBlue-1 with NovaBlue Singles Competent Cells (Millipore-Sigma). The expression constructs were then transformed into Tuner (DE3)pLacI (Millipore-Sigma), a derivation of BL21 strain, for inducible over-expression. Fusion proteins were induced with a standard IPTG method for 3 h and crude bacterial lysate was prepared according to manufacturer's protocol. The GST pulldown assay was performed with Pierce GST Protein Interaction Pull-Down Kit (ThermoFisher Scientific) using GST-BRCT and either WT or mutant 6His-p50 as indicated. GST fusion proteins were immobilized on glutathione-Sepharose beads and after washing the interacting complex was eluted and analyzed on SDS–PAGE followed by immunoblotting.

**In vitro kinase assay**. Fifty nanograms of bacterially expressed wild type or mutant p50 was incubated with 50 ng active recombinant CHK1 (Active Motif), 50 mM ATP, and 5 mCi of [γ-³²P]ATP in kinase buffer (50 mM Tris-HCl pH 7.5, 1 mM DTT, 10 mM MgCl₂, 10 mM MnCl₂) for 30 min at 30 °C. Following SDS–PAGE, membranes were analyzed by autoradiography and by immunoblot with the indicated antibody after addition of SDS sample buffer.

**In vitro ubiquitination assay**. This was performed using 6HIS-p50^wt or 6HIS-p50^DE2A, GST-FL-BARD1 (full length) purified from *E. coli* with Pierce glutathione magnetic agarose beads (Life Technologies) and recombinant human BRCA1 (Abcam). Ubiquitination assays were performed, as previously described[64], with 400 ng ubiquitin-activating enzyme (UBE1) (Boston Biochem), 400 ng purified Ubc5c (Boston Biochem), 250 ng purified GST-FL-BARD1, 500 ng recombinant human BRCA1, 2 µg Myc-ubiquitin (Boston Biochem) and 1 µg His-p50^wt or HIS-p50^DE2A in reaction buffer containing 50 mM Tris (pH 7.6), 5 mM MgCl₂, 2 mM ATP, and 2 mM DTT. Reactions were carried out at 37 °C for 60 min, then stopped by boiling in SDS sample buffer.

**Quantification of blots**. Semi-quantitative analysis of Co-IPs, IBs and protein stability assays in the presence of CHX was performed using ImageJ (v1.52a) gel analysis tool set, normalized to loading. For stability, the degradation model and half-life were fit and calculated with Prism Graphpad (v7). Also, semi-quantitative analysis of S337 phosphorylation following TAM treatment was fit with Prism

Graphpad (v7). All graphical quantification data are representative of at least two separate independent biological experiments.

**Lentiviral production and infection.** Recombinant lentiviral particles were produced by transfecting 293T cells with the lentiviral expression plasmid pLVX carrying the appropriate p50 construct and the packaging plasmids, psPAX2 and pMD2.G, using *Trans*IT-LT1 Transfection Reagent (Mirus Bio). In all, $8 \times 10^5$ cells were cultured in 10-cm plates with Opti-MEM (ThermoFisher Scientific) and transfection performed when the cell density reached 50–60% confluency. After 6 h, the culture medium was replaced with fresh DMEM/10% FBS (Atlanta Biologicals) and 48 h later the medium was collected and centrifuged at $4000 \times g$ at 4 °C for 10 min to remove cellular debris. The supernatant was filtered through a 0.45 μm filter and then concentrated with Lenti-X concentrator (Clontech). For lentivirus infection, target cells were seeded in 10-cm plates at 50% confluency and infected with lentiviral particles at multiplicity of infection (M.O.I.) of 5, in the presence of 8 μg/ml polybrene (Sigma-Aldrich). The cells were then selected with 2 μg/ml puromycin (Sigma-Aldrich) for 7 days prior to use.

**Mouse xenograft experiments.** Six- to 7-week-old male nude mice (Hsd:Athymic Nude-*Foxn1nu*) purchased from Harlan-Envigo were used in accordance with guidelines of the Institutional Animal Care and Use Committee of the University of Chicago. Immortal and transformed *Nfkb1*$^{-/-}$ MEFs from the same passage were infected with either p50$^{wt}$ or p50$^{DE2A}$ and stable cells selected. Subsequently, one million ($10^6$) cells were suspended in 100 μl of sterile 1xPBS and mixed with 100 μl of Matrigel (BD Biosciences, 354263). The mixture was injected subcutaneously into the right flank of nude mice. Mice were sacrificed at two weeks and tumor dimensions measured according to a standard protocol. Tumor volume was calculated with the formula, $V = \pi/6 \times L \times W \times H$. The experiment was repeated with a distinct batch of *Nfkb1*$^{-/-}$ MEFs to minimize the chance of a clone-specific effect. Tumors were harvested and lysates used for IB.

**Neuroblastoma tissue, breast cancer tissue microarray, and immunohistochemistry.** For neuroblastoma specimens, patients who presented to the University of Chicago with a diagnosis of neuroblastoma between 2006–2017 were retrospectively enrolled following Institutional Review Board (IRB) approval at the University of Chicago. Among these patients, 25 were deemed to have adequate tissue available for immunohistochemical (IHC) staining after evaluation by a neuropathologist. For IHC analysis, sections were cut at 4 μm thickness and for all studies, serial sections were used. Slides were stained using Leica Bond RX automatic stainer. After addition of epitope retrieval solution I (Leica Biosystems, AR9961) for 20 min, primary antibody was applied for 1 h. The primary antibodies used were: anti-BARD1 (sc-74559 Santa Cruz, 1:200 dilution), anti-p50 (ab32360 Abcam, 1:200 dilution) and anti-PHOX2B (ab183741 Abcam, 1:1000, prepared fresh). Subsequently, antigen-antibody binding was detected with Bond polymer refine detection (Leica Biosystems, DS9800). Sections were stained with hematoxylin and covered with cover glasses.

For breast cancer tissue microarray (TMA) construction, formalin-fixed paraffin-embedded (FFPE) tissue was obtained from the surgical pathology archive of the University of Chicago after IRB approval. TMAs were constructed by placing FFPE cores (both tumor and adjacent normal tissue) into a recipient paraffin block using an automated arrayer (ATA-27, Beecher Instruments, Silver Spring, MD). Tissue samples were de-identified. MCF-7 and HeLa cells were also prepared as cell blocks and arrayed into a TMA blocks as controls. Slides were stained with Hematoxylin and Eosin (H&E) and evaluated for morphological quality by a trained pathologist. IHC of TMAs was performed as for neuroblastoma slides except no PHOX2B staining was performed. A total of 372 cores on three separate TMAs were assessed and contained tissue from 91 patients. Of these, 86 patients had adequate tissue for IHC analysis. Tumors incorporated into TMAs included a variety of breast cancer pathologies, including luminal A and B tumors, ductal carcinoma in-situ (DCIS) specimens, and lymph node metastases. Survival status for breast cancer patients (alive or dead) was determined at the time each TMA was constructed. Patients placed in each TMA were diagnosed at approximately the same time and assessment of survival determined 16 years after diagnosis. Two patients were excluded from survival analysis because of uncertain diagnosis date (one patient had low BARD1/p50 staining and one had high BARD1/p50).

IHC staining of both neuroblastoma tissue and breast cancer TMAs was scored in a blinded fashion by two observers and inter-observer differences verified by re-examination. Importantly, only nuclear staining of either BARD1 or p50 was considered positive. This was especially important with anti-p50 IHC as this antibody also recognizes the cytoplasmic parental protein, p105. In addition, in neuroblastoma tissue, only cells that stained positive for PHOX2B were considered in the evaluation of BARD1 and p50 as cells negative for PHOX2B are not neoplastic. For neuroblastoma, staining was scored in a semi-quantitative fashion based on a four-tier system: 0 (no staining), 1 (<25% positive cells), 2 (25–75% positive), and 3 (>75% positive). This score was converted into a binary grade where a score of 0 or 1 was deemed low and a score of 2 or 3 deemed high. For breast cancer tissue, scoring was performed based on both staining proportion and staining intensity as described for breast tissue[46]. Again, the overall score was

dichotomized as either high or low. Examples of each category are shown (Fig. 7 and Supplementary Fig. 7).

**Statistics and reproducibility.** Statistical analyses were performed with two-sided unpaired Student's *t*-test for single comparison. *P*-values < 0.05 were considered statistically significant. Experimental data points were expressed as means ± SEM. For patient samples, Fisher's exact test (both two-sided and one-side, where indicated) was used. For Box plot: center-line is the median, box limits represent upper and lower quartiles and whiskers are minimum and maximum. To ensure reproducibility, blots were repeated at least twice as indicated in the specific methods and legends. For IHC, antibodies were initially titrated to identify the optimal dilution. Subsequently, validation of staining was performed by independently staining the first specimen twice at the optimal dilution.

**Reporting summary.** Further information on research design is available in the Nature Research Reporting Summary linked to this article.

## Data availability

ChIP-Seq data have been deposited at the NCBI GEO repository under accession number: GSE129618. All other data can be provided by the corresponding author upon reasonable request. Source data are provided with this paper.

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

## Acknowledgements
We are grateful to Richard Baer for BARD1 constructs and *Bard1*-null mouse mammary carcinoma cells, Phillip Connell for HCC1937 cells, Susan Cohn for neuroblastoma cell lines, Oscar Fernandez-Capetillo for plasmids, Elizabeth Davis for assistance with metaphase spreads, Peter Pytel for assistance with neuroblastoma tissue procurement and analysis and Zhongqin Zhang for technical assistance with MS/MS. This work was supported by National Institutes of Health (NIH) grant R01CA136937 to B.Y. and the Ludwig Center for Metastasis Research.

## Author contributions
L.W. designed and performed the majority of the experiments, analyzed the data, and helped write the manuscript. C.D.C. and A.G. performed gel shift and metaphase spread experiments. G.F.K. and O.I.O. provided breast cancer TMAs and helped with scoring of breast cancer cores. J.N. and A.-P.C. helped with neuroblastoma studies. R.R.W. provided resources and reviewed the manuscript. B.Y. supervised the entire project, analyzed the data, and wrote the manuscript. All authors approved the final manuscript prior to submission.

## Competing interests
The authors declare no competing interests.
