## [Peer Review File · Nature Communications]

Reviewers' comments:

Reviewer #1 (Remarks to the Author):

Wu et al. described in this paper that p50 mono-ubiquitination by directly interacting with BARD1 BRCT domains via a C-terminal phospho-serine motif regulates S phase progression and maintains genome integrity. The authors demonstrated that p50 phospho-serine motif interacts with BARD1 BCRT domain in response to ATR-CHK1 activation. Interaction of BARD1 with p50 promotes p50 mono-ubiquitination at K354/K356, leading to increase of p50 stability and nuclear to cytoplasmic shuttling of p50. Furthermore, they demonstrated that p50 interaction with BARD1 and p50 mono-ubiquitination acts to slow S phase progression by decreasing the recruitment of p50 to the CCNE1 promoter in S phase relative to G1. Finally, they showed that p50 mono-ubiquitination is required for maintaining genome stability and found that p50 protein level correlates with BARD1 in human cancer specimens. Overall, this study is interesting but raising several questions unanswered. There are some concerns regarding several key points:

Comments:

1. In Fig 2e, both S328 and S337 at p50 can be phosphorylated by CHK1. It was shown that phosphorylation of p50 at S337 is required for interacting with BDAD1 and mono-ubiquitination of p50. However, the authors initially used affinity purification of HA-tagged wildtype p50 (p50WT) or an S328A mutant (p50S328A) to identify BARD1. What is the relationship between phosphorylation of p50 at S328 and S337 for maintaining genome integrity?
2. In Fig 3, ATR-mediated p50 mono-ubiquitination is BARD1 dependent through interacting with p50 phospho-serine motif and is required for maintaining genome integrity. However, poly-ubiquitination of p50 can also be regulated by ATR and BARD1. What is the function of poly-ubiquitination of p50?
3. IN Fig 6 a-b, the maximal phosphorylation level of p50 at S337 and the maximal expression level of BARD1 are at early S phase. Why mono- ubiquitination of p50 occurs mainly at S phase? Why the expression level of cyclin E is high at S phase? How does p50 mono- ubiquitination regulate cyclin E expression?
4. During cell cycle, it may be better to show which protein involves in dephosphorylation of p50 or de-mono-ubiquitination of p50 to complete the story.
5. In Fig 6f, it is not clear whether cyclin E expression levels and chromosome breaks are higher in p50 DE2A tumor than in p50 WT tumor.
6. In Figure 7a, loss of p50 mono-ubiquitination induces genomic instability. In addition, mono-ubiquitination of p50 can maintain its stability and increase nuclear export ability. Why the authors detected more clinical tumor specimens with high BRAD1 and high nuclear p50? What is the clinical outcome of progression survival or overall survival for patients with the specimens with high BARD1 and high p50 expression levels?

Reviewer #2 (Remarks to the Author):

In this manuscript, Wu et al. report that ATR activation, resulting from either S-phase entry or genotoxic stress, induces CHK1-mediated phosphorylation of p50 at residue S337. Interestingly, this

modification creates a binding site for the BRCT domain of BARD1, a tumor suppressor and subunit of the BRCA1/BARD1 heterodimer. In a series of well-executed experiments, the authors show that the p50/BARD1 interaction leads to monoubiquitination, stabilization and cytoplasmic relocation of p50. As a consequence, p50 binding to chromatin, including the CCNE (cyclin E1) gene, is reduced, allowing CCNE de-repression, S-phase progression, and genome instability. These findings are consistent with previous evidence that p50 can function in a manner that suppresses tumor development and they provide a firm molecular basis for interplay between the NFkB and BRCA1 pathways. As such this is potentially an important contribution. However, by not properly addressing the role of BRCA1 in this process, the authors leave the impression (likely incorrect) that the p50/BARD1 interaction represents a mysterious “BRCA1-independent” function of BARD1. Also, the identity of the E3 ligase that mediates p50 monoubiquitination is not addressed; is it the BRCA1/BARD1 heterodimer, BARD1 alone (unlikely), or a distinct E3 ligase functioning downstream of BARD1?

Specific comments:

1. On page 4, the authors state that “BARD1 also has BRCA1-independent actions such as promoting damage-induced apoptosis”. This notion, which was promulgated in a couple of review articles from one lab, originally emerged from experiments that entailed non-physiological levels of BARD1 overexpression. However, subsequent studies using clean genetics systems do not support this idea. For example, the phenotypes of *Brca1*- and *Bard1*-null mice are essentially indistinguishable, whether gene inactivation is germline, resulting in embryonic lethality, or induced in mammary epithelial cells, resulting in tumor formation. Indeed, BRCA1 and BARD1 are epistatic for essentially all activities tested to date in studies conducted in human, mouse, frog, worm, and rice cells. Moreover, the two genes co-exist evolutionarily, such that both are either present (e.g., in mammals, frogs, worms, and rice) or absent (e.g., yeast and drosophila) together. In vivo, most, if not all, cellular BRCA1 polypeptides are associated with BARD1, and vice versa. Thus, although we cannot formally exclude the possibility that in some unique settings BARD1 exerts some activity independent of BRCA1, it is likely to be a very minor component of BARD1 function.
2. On page 11, while discussing BARD1-mediated monoubiquitination of p50, the authors state “that while BRCA1 is not necessary for this process, BRCA1 enhances the effect.” However, the conclusion that BRCA1 is not necessary for this process is based solely on this one experiment, and it implies that the residual mono-ubiquitinated bands that appear in lanes 3 and 5 of Figure 3e are generated by the E3 activity of BARD1 alone. However, this is inconsistent with substantial existing evidence. For examples, early studies showed that BRCA1 alone displays a modest E3 activity in vitro, which is dramatically enhanced by the presence of BARD1 and formation of the BRCA1/BARD1 heterodimer. However, in those cases in which it was tested, BARD1 alone was devoid of detectable E3 activity. These results were subsequently rationalized by structural studies, which showed that the RING domain of BRCA1 contains the conventional alpha helix that RING E3 proteins use to bind their cognate E2 ubiquitin conjugases. However, this alpha helix is conspicuously absent from the structure of the BARD1 RING domain. Thus, the RING domain of BRCA1, but not BARD1, has the ability to bind E2 conjugases on its own. Given these findings, it would seem unlikely that p50 monoubiquitination is mediated by BARD1 alone.
3. In light of these uncertainties, the authors could provide more convincing cellular data to distinguish whether p50 monoubiquitination is or is not dependent on BRCA1.
4. To further establish the identity of the responsible E3 ligase, the authors could examine whether

p50 can be monoubiquitinated in vitro by purified preparations of BARD1 alone, BRCA1 alone, or the BRCA1/BARD1 heterodimer.

5. Figure 1c, right panels: what is the high MW species that appears in both the BARD1 and IgG immunoprecipitates?

6. Figure 1e: much of the key control band of the GFP-ER immunoprecipitate is missing. It might be better to repeat this experiment to exclude technical problems, such as poor immunoblot transfer.

7. In the S phase of undamaged cells, is p50-S337 phosphorylation also mediated by CHK1?

8. The ability of BARD1 to bind a phosphorylated p50 is very intriguing, suggesting that the BRCT domain of BARD1, like that of NBS1, has the potential to recognize both PAR and phospho-protein ligands. However, to confirm the phospho-dependent nature of the p50/BARD1 interaction, the authors should conduct two key experiments that are conventionally used to establish the validity of BRCT/phosphoprotein interactions: is the p50/BARD1 interaction impaired by 1) phosphatase treatment and 2) by mutations of the structurally-defined BRCT phospho-recognition residues (such as BARD1- S575)?

Reviewer #3 (Remarks to the Author):

The authors identified BARD1, a prominent BRCA1-associated protein, to interact with the p50 subunit of NF- κ B based on mass spectrometric analysis of transfected 293T cells (Fig. 1). The site of interaction involved p50S238, a phosphorylation site previously implicated in maintenance of genome stability. Co-transfection assays into 293T cells were used to further characterize interactions between these proteins (Figs. 2-4). These studies implicated p50S337 and the BRCT domain of BARD1 via ATR-induced Chk1 kinase activation. The RING1 domain of BARD1 was shown to mono-ubiquitinate p50 at K354 and K356. Functional effects of p50/BARD1 interaction were tested in different ways. BARD1 interaction stabilized p50 protein levels and mono-ubiquitination resulted in cytoplasmic relocation of p50 from the nucleus (Fig. 5). Cytosolic localization was accompanied by reduced p50 binding to chromatin. Cell cycle progression of mouse embryo fibroblasts (MEFs) was affected by disrupting p50/BARD1 interactions with point mutations in either protein (Fig. 6). Introduction of WT p50, but not a mutated version that cannot be mono-ubiquitinated, reduced genomic instability in p50-deficient MEFs (Fig. 7). Finally, p50 and BARD1 levels were statistically correlated in a variety of human tumor samples (Fig. 7), suggesting that this interaction regulated genomic instability during disease.

The authors have done a comprehensive analysis of p50/BARD1 interactions (Figs. 1-4). The data is clean, however there is little quantitation of the co-IPs, GST-pull downs or phosphorylation states. Additionally, virtually all the biochemistry is carried out under over-expression conditions in co-transfected 293T cells. While this is a good place to start, additional quantitative studies utilizing endogenous proteins is required. Functional studies point to a role for p50 stabilization by BARD1 that results in accelerated cell cycle progression via cyclinE1 up-regulation. A strong point here is the comparison between 2KR and 2KR-Ub mutants for cell cycle studies. However, the effects of these mutants were not compared with WT p50 in the same assay (Fig. 6g, h). I am also circumspect about p50 stability in the presence or absence of BARD1 which was not quantified, as well use of different cell lines to measure effects on the cell cycle.

Overall, this study identifies a plausible mechanism by which NF- κ B p50 confers genomic stability.

This is an important question and I believe that the insight of a ATR-driven association of p50 with BARD1 and associated downstream effects is novel and worthy of consideration in Nature Communications. The major shortcoming in the current version of the manuscript is reliance on over-expression systems to demonstrate association and cellular effects. In my opinion, a more focused manuscript that connects DNA damage signals via endogenous p50/BARD1 to a cell cycle outcome in an untransformed cell type (e.g. MEFs) would greatly strengthen the impact of the study. Such an analysis could be added to the present data after removing much of Figs. 1-4 to supporting material. The specifics of re-organization can be discussed between the authors and editors if this option is taken.

Response to Referees

We would like to thank all the Reviewers for their comments and believe that the revisions they recommended have significantly improved the overall impact and clarity of the manuscript. We have endeavored to address every concern raised.

In response to the **Editor's** comments, we have attempted to address the concerns regarding the role of p50 phosphorylation and the mechanism and effect of p50 ubiquitination. Also, we addressed the general concerns regarding the need for examination of endogenous proteins in an untransformed cell line. The specific critiques of each Reviewer are given in black and followed by our response in red in a point-by-point fashion.

Reviewer #1

Wu et al. described in this paper that p50 mono-ubiquitination by directly interacting with BARD1 BRCT domains via a C-terminal phospho-serine motif regulates S phase progression and maintains genome integrity. The authors demonstrated that p50 phospho-serine motif interacts with BARD1 BCRT domain in response to ATR-CHK1 activation. Interaction of BARD1 with p50 promotes p50 mono-ubiquitination at K354/K356, leading to increase of p50 stability and nuclear to cytoplasmic shuttling of p50. Furthermore, they demonstrated that p50 interaction with BARD1 and p50 mono-ubiquitination acts to slow S phase progression by decreasing the recruitment of p50 to the CCNE1 promoter in S phase relative to G1. Finally, they showed that p50 mono-ubiquitination is required for maintaining genome stability and found that p50 protein level correlates with BARD1 in human cancer specimens. Overall, this study is interesting but raising several questions unanswered. There are some concerns regarding several key points:

Comments:

1. In Fig 2e, both S328 and S337 at p50 can be phosphorylated by CHK1. It was shown that phosphorylation of p50 at S337 is required for interacting with BDAD1 and mono-ubiquitination of p50. However, the authors initially used affinity purification of HA-tagged wildtype p50 (p50WT) or an S328A mutant (p50S328A) to identify BARD1. What is the relationship between phosphorylation of p50 at S328 and S337 for maintaining genome integrity?

Although we incorporated p50-S328A into our initial affinity purification experiments, once BARD1 was identified as a p50 interacting partner and S337 determined to be part of the BARD1 interaction motif, we pursued the role of S337 in this pathway and not S328.

With regard to S328, we previously examined the relationship between this residue and genome integrity and reported that S328 (S329 in the prior report) is required for genome integrity (Crawley CD, et al. *Cell Cycle*, 2015).

As far as the relationship between S337 and genome integrity is concerned, we initially made mutants of this residue planning to examine genome integrity. However, we found that p50-S337A had substantially reduced DNA binding compared to WT and

that S337D had almost no DNA binding (**Supplementary Fig 4d**), findings that were also noted by others (Hou S, *et. al. JBC*, 278, 2003). As p50-S337A bound some DNA sequences, we examined this on a genome-wide scale and found that S337A had drastically reduced chromatin recruitment relative to WT (the data are not in the paper, but we include it here for the Reviewer- see **FIGURE**). Given that p50 mediates its effects by DNA binding, the lack of binding by S337 mutants indicated that this residue is essential for p50 function and that its mutation would behave like a p50 null (i.e. empty vector). These observations suggested that examining S337 mutants would be significantly less informative of the BARD1/p50 pathway than examining the ubiquitin-site mutant, p50-2KR, which *is* recruited to chromatin (**Supplementary Fig 5d**). We therefore examined genome integrity using p50-2KR and found that the mono-ubiquitination sites were required for genome integrity. Given that S337 phosphorylation is required for p50 mono-ubiquitination, we conclude that S337 phosphorylation is required for genome integrity.

FIGURE. ChIP-Seq analysis of p50-WT and p50-S337A. Average peak intensity demonstrates drastically reduced chromatin binding in cells expressing S337A compared to WT (data from triplicate samples pooled and repeated with similar results).

Finally, although the exact relationship between S328 and S337 phosphorylation is unclear, we examined whether S337 can be phosphorylated in the presence of p50-S328A. Notably, loss of S328 phosphorylation by mutation did not block S337 phosphorylation in response to replication stress (**Supplementary Fig. 2c**).

2. In Fig 3, ATR-mediated p50 mono-ubiquitination is BARD1 dependent through interacting with p50 phospho-serine motif and is required for maintaining genome integrity. However, poly-ubiquitination of p50 can also be regulated by ATR and BARD1. What is the function of poly-ubiquitination of p50?

We appreciate the Reviewer's interest in understanding the function of p50 poly-ubiquitination. However, as we have not identified the poly-ubiquitination sites, it is not yet possible for us to say what the exact function of p50 poly-ubiquitination is. Also, it is likely that an unidentified ligase, possibly induced by BARD1 or ATR, is involved in p50 poly-ubiquitination. While identification of such a ligase might help in understanding the role of p50 poly-ubiquitination, this undertaking is beyond the scope of the current work. In addition, although it is true that stimulation of ATR or over-expression of BARD1 promoted both mono-ubiquitination and poly-ubiquitination (Figs 3a and c), loss of the interaction between p50 and BARD1 (or loss of the ligase ability of BARD1) specifically affected mono-ubiquitination, not poly-ubiquitination (Figs 3e, 3f and 3g). This finding underlines our focus specifically on mono-ubiquitination in the BARD1/p50 interaction.

Ultimately, given that the function of poly-ubiquitination is dependent on the type of ubiquitin link formed (K48 linked chains promote proteasomal degradation whereas K63 chains are associated with cellular shuttling and signaling), it is highly likely that poly-ubiquitination acts to modulate the metabolism and/or cellular distribution of p50.

3. IN Fig 6 a-b, the maximal phosphorylation level of p50 at S337 and the maximal expression level of BARD1 are at early S phase. Why mono- ubiquitination of p50 occurs mainly at S phase? Why the expression level of cyclin E is high at S phase? How does p50 mono-ubiquitination regulate cyclin E expression?

The Reviewer is correct in noting that maximal p50 phosphorylation is at early S phase. However, we do not show maximal BARD1 expression at this time. We specifically demonstrate maximal BARD1/p50 interaction at this time (Figs 6b and c upper blots are Co-IP analyses).

Answers to the specific questions raised are below:

a) Why mono- ubiquitination of p50 occurs mainly at S phase?

p50 mono-ubiquitination occurs mainly during S phase because i) BARD1 mono-ubiquitinates p50 and ii) BARD1 interacts with p50 specifically during S phase. In addition, as BARD1 requires BRCA1 to mono-ubiquitinate p50, the finding that mono-ubiquitination occurs in S phase is consistent with the prior report that BARD1 associates with nuclear BRCA1 specifically in S phase (Jin Y, *et. al. PNAS*, 94, 1997).

b) Why the expression level of cyclin E is high at S phase?

We presume the Reviewer is asking this because Cyclin E regulates the G1/S transition and therefore its expression would be highest during this transition. While Cyclin E protein increases at the G1/S transition, it is also elevated during S phase (one reference for this: Moroy and Geisen 2004, *IJBCB*). In our experiments, cells were synchronized and released, and the specific phases of the cell cycle determined by flow cytometry. Early S phase (i.e. the G1/S transition) was 2 hours after release, S phase was at 4 hours and G2 was 8 hours after release. There is no Cyclin E in the quiescent state (G1). Subsequently, Cyclin E increased and was seen at the 2 hour time point (early S). However, the half-life of Cyclin E is 4 - 5 hours (Kossatz U, *et. al. JCI*, 2010, McEvoy JD, *et. al. Mol Cell Biol*, 2007), therefore the increase in Cyclin E protein at the G1/S transition would still be high 2 hours later (i.e. S phase in our protocol). In addition, while the cell cycle phases documented represent the times when the greatest number of cells were in the indicated phase, even at the 4 hour time point, a certain percentage of cells were still in the early stage of S phase (i.e. the G1/S transition). These observations explain why we see high Cyclin E in S phase. Importantly, there was very little Cyclin E protein in G1 and G2 (the times with the fewest cells transitioning into S). Moreover, these findings were seen in more than one cell line and are grossly in line with the role of Cyclin E in regulating G1/S transition.

c) How does p50 mono- ubiquitination regulate cyclin E expression?

In our model, constitutively produced p50 dimers bind the *CCNE* promoter in G1 and maintain Cyclin E expression at a low level. On S phase transition, S337 phosphorylation leads to decreased p50 binding to the *CCNE* promoter

facilitating Cyclin E expression and S phase progression. p50 mono-ubiquitination occurs as a result of S337 phosphorylation. From a mechanistic standpoint, mono-ubiquitination promotes the stabilization and nuclear export of p50. We propose that this stabilization maintains p50 in the phosphorylated state preventing it from re-binding the CCNE promoter. While S337 phosphorylation blocks p50 DNA binding, whether mono-ubiquitination also alters binding is not as clear (the ubiquitin-site mutant, p50-2KR, does have altered chromatin recruitment relative to WT, Supplementary Fig. 5d). In addition, we hypothesize that nuclear export further contributes to preventing mono-ubiquitinated p50 from binding promoters. The presence of the mono-ubiquitination sites close to the p50 nuclear localization sequence (NLS) likely plays a role in this nuclear export. Ultimately, p50 that cannot be mono-ubiquitinated (i.e. p50-2KR) cannot regulate Cyclin E like WT, resulting in deregulated S phase and genome instability. Although p50-2KR is primarily nuclear, it is less stable than WT potentially explaining why it does not inhibit Cyclin E like WT-p50 (Fig. 6l).

4. During cell cycle, it may be better to show which protein involves in dephosphorylation of p50 or de-mono-ubiquitination of p50 to complete the story.

Identifying the phosphatase or de-ubiquitinating enzyme (DUB) that de-phosphorylates or de-mono-ubiquitinates p50 is something we have been actively working on. As of now, we have not yet identified the specific enzymes involved. While identification of these proteins would add to the mechanistic understanding of this pathway, their identification would constitute a significant undertaking, involving potentially a large-scale screen followed by mechanistic analysis. Such experiments would result in new manuscripts, one centered on the phosphatase and another on the DUB. As such, identifying these proteins is somewhat beyond the scope of the current study.

5. In Fig 6f, it is not clear whether cyclin E expression levels and chromosome breaks are higher in p50 DE2A tumor than in p50 WT tumor.

To address this we first examined cyclin E. This data is now presented as (**Supplementary Fig. 6e**) and demonstrates by immunoblot that there is higher cyclin E abundance in p50 DE2A compared to p50 WT. Examination of chromosomal breaks was problematic because: i) the immortal *Nfkb1*^{-/-} MEFs are transformed and therefore have numerous basal chromosomal breaks and ii) analysis of breaks in tumors was difficult as very few metaphase nuclei (required for chromosomal analysis) are visible in tumor sections. We did look at DNA damage by analysis of γ H2AX staining and noted no difference in this marker between MEFs expressing p50 WT and p50 DE2A.

6. In Figure 7a, loss of p50 mono-ubiquitination induces genomic instability. In addition, mono-ubiquitination of p50 can maintain its stability and increase nuclear export ability. Why the authors detected more clinical tumor specimens with high BRAD1 and high nuclear p50?

What is the clinical outcome of progression survival or overall survival for patients with the specimens with high BARD1 and high p50 expression levels?

The first point we would like to note is that when we say 'high' this equates with what would be considered a normal amount of protein, i.e. that seen in non-malignant cells. It is the presence of low BARD1 or low p50 that is unusual. As far as *why* more tumors have high levels of these proteins, we give the following explanation. In breast cancer, <1% have *BARD1* mutations, and although *BRCA1* stabilizes BARD1, only up to 10% of these tumors have *BRCA1* mutations. These findings suggest that only about 10% of breast cancers would have low BARD1 underlining why more tumors have high BARD1 protein. For p50, it is important to appreciate that only a fraction of nuclear p50 is mono-ubiquitinated and specifically only in cells during S phase (i.e. in tissue samples, the majority of nuclear p50 is **not** phosphorylated/ubiquitinated). Therefore, even in the presence of high (normal) BARD1, only some of the nuclear p50 would be exported out of the nucleus leaving the majority of cells with high amount of nuclear p50. On the other hand, as p50 is stabilized by BARD1, in cells with low nuclear BARD1 (about 10% of breast cancers), p50 protein would not be stabilized and its overall level reduced. These observations explain why most breast cancers would be expected to have high BARD1 and high p50 in the nucleus.

With regard to neuroblastoma, the same reasoning holds for p50 protein. In the case of BARD1, only 'high-risk' neuroblastoma has been associated with *BARD1* and altered BARD1 expression is only seen in a small percentage of these high-risk tumors. In our cohort, 20 of 25 patients were high-risk. Based on the above reasoning, it is expected that the majority of our neuroblastoma specimens would have normal (high) levels of BARD1 protein.

To answer the second question, we obtained survival data. For breast cancer, the only data available was whether the patients were dead or alive at the time each TMA was constructed. Importantly, in each individual TMA, the diagnosis for all the patients was made at approximately the same time, and OS determined 16 years after diagnosis. We correlated the IHC staining score for BARD1 and p50 with whether patients were alive or dead at the time of survival analysis. Remarkably, for p50, a significant correlation between nuclear staining and vital status was seen ($P = 0.048$, Fisher exact test). This is now included as **Supplementary Fig. 7g**. Specifically, patients who had low nuclear p50 did significantly worse than those with high staining. For BARD1, only 6 patients with low BARD1 staining had survival data, no significant correlation was evident likely due to lack of sufficient numbers.

For neuroblastoma, although we obtained both OS and progression free survival (PFS) data, given the low total patient number, we were unable to obtain meaningful statistics and no correlation with survival was seen. Specifically, with BARD1, of the 4 patients with low BARD1, 2 were dead and 2 alive, with both living patients having no progression 6 years after diagnosis. For p50, of the 7 patients with low p50, 3 were dead and 4 alive. Those with high BARD1 and high p50 had similar mixed OS and PFS.

Reviewer #2

In this manuscript, Wu et al. report that ATR activation, resulting from either S-phase entry or genotoxic stress, induces CHK1-mediated phosphorylation of p50 at residue S337. Interestingly, this modification creates a binding site for the BRCT domain of BARD1, a tumor suppressor and subunit of the BRCA1/BARD1 heterodimer. In a series of well-executed experiments, the authors show that the p50/BARD1 interaction leads to monoubiquitination, stabilization and cytoplasmic relocation of p50. As a consequence, p50 binding to chromatin, including the CCNE (cyclin E1) gene, is reduced, allowing CCNE de-repression, S-phase progression, and genome instability. These findings are consistent with previous evidence that p50 can function in a manner that suppresses tumor development and they provide a firm molecular basis for interplay between the NFkB and BRCA1 pathways. As such this is potentially an important contribution. However, by not properly addressing the role of BRCA1 in this process, the authors leave the impression (likely incorrect) that the p50/BARD1 interaction represents a mysterious “BRCA1-independent” function of BARD1. Also, the identity of the E3 ligase that mediates p50 monoubiquitination is not addressed; is it the BRCA1/BARD1 heterodimer, BARD1 alone (unlikely), or a distinct E3 ligase functioning downstream of BARD1?

Specific comments:

1. On page 4, the authors state that “BARD1 also has BRCA1-independent actions such as promoting damage-induced apoptosis”. This notion, which was promulgated in a couple of review articles from one lab, originally emerged from experiments that entailed non-physiological levels of BARD1 overexpression. However, subsequent studies using clean genetics systems do not support this idea. For example, the phenotypes of Brca1- and Bard1-null mice are essentially indistinguishable, whether gene inactivation is germline, resulting in embryonic lethality, or induced in mammary epithelial cells, resulting in tumor formation. Indeed, BRCA1 and BARD1 are epistatic for essentially all activities tested to date in studies conducted in human, mouse, frog, worm, and rice cells. Moreover, the two genes co-exist evolutionarily, such that both are either present (e.g., in mammals, frogs, worms, and rice) or absent (e.g., yeast and drosophila) together. In vivo, most, if not all, cellular BRCA1 polypeptides are associated with BARD1, and vice versa. Thus, although we cannot formally exclude the possibility that in some unique settings BARD1 exerts some activity independent of BRCA1, it is likely to be a very minor component of BARD1 function.

We greatly appreciate the Reviewer’s extensive comments regarding this topic (it has not been properly addressed in the literature). We agree with the notion that the Ub-ligase activity derives from the heterodimer of BRCA1/BARD1. We have adjusted our text to remove any indications of BRCA1-independent actions of BARD1.

2. On page 11, while discussing BARD1-mediated monoubiquitination of p50, the authors state “that while BRCA1 is not necessary for this process, BRCA1 enhances the effect.”

However, the conclusion that BRCA1 is not necessary for this process is based solely on this one experiment, and it implies that the residual mono-ubiquitinated bands that appear in lanes 3 and 5 of Figure 3e are generated by the E3 activity of BARD1 alone. However, this is inconsistent with substantial existing evidence. For examples, early studies showed that BRCA1 alone displays a modest E3 activity in vitro, which is dramatically enhanced by the presence of BARD1 and formation of the BRCA1/BARD1 heterodimer. However, in those cases in which it was tested, BARD1 alone was devoid of detectable E3 activity. These results were subsequently rationalized by structural studies, which showed that the RING domain of BRCA1 contains the conventional alpha helix that RING E3 proteins use to bind their cognate E2 ubiquitin conjugases. However, this alpha helix is conspicuously absent from the structure of the BARD1 RING domain. Thus, the RING domain of BRCA1, but not BARD1, has the ability to bind E2 conjugases on its own. Given these findings, it would seem unlikely that p50 monoubiquitination is mediated by BARD1 alone.

Again we agree with the Reviewer regarding this point. To address this, we have removed the comment in the text (page 11) and have now added new experiments demonstrating that BARD1 and BRCA1 are both required for efficient mono-ubiquitination of p50. First, using the BRCA1-deficient cell line, HCC1937, we show that in these cells p50 mono-ubiquitination only occurs when BRCA1 is added (**Fig. 3h**) a finding blocked when *BARD1* is knocked down (**Fig. 3h**). Subsequently, in an independent cell line (*Nfkb1*^{-/-} MEFs), we show that knockdown of either *Brca1* or *Bard1* blocked p50 mono-ubiquitination (**Supplementary Fig. 3e**). Importantly, in this experiment to counter destabilization of *Brca1* or *Bard1* by depletion of the other protein, we also over-expressed each subunit. Finally, using *in vitro* ubiquitination, we see that p50 is mono-ubiquitinated only in the presence of both BRCA1 and BARD1 (**Fig. 3i**). These results clearly demonstrate that both BARD1 and BRCA1 are required for p50 mono-ubiquitination.

3. In light of these uncertainties, the authors could provide more convincing cellular data to distinguish whether p50 monoubiquitination is or is not dependent on BRCA1.

We addressed this question in the answer above. Specifically, we used BRCA1-deficient human cells, HCC1937, and demonstrated the requirement of both BRCA1 and BARD1 for p50 mono-ubiquitination (**Fig. 3h**). Subsequently, in mouse cells (MEFs), we further verified the requirement of both *Brca1* and *Bard1* for mono-ubiquitination (**Supplementary Fig. 3e**). Finally, we demonstrated the requirement of BRCA1 using *in vitro* ubiquitination analysis (**Fig. 3i**). Together, these data indicate that p50 mono-ubiquitination *is* dependent on BRCA1.

4. To further establish the identity of the responsible E3 ligase, the authors could examine whether p50 can be monoubiquitinated in vitro by purified preparations of BARD1 alone, BRCA1 alone, or the BRCA1/BARD1 heterodimer.

As requested, we performed *in vitro* ubiquitination using purified BARD1 and BRCA1 protein. These data, presented in **Fig. 3i**, indicate that both BARD1 and BRCA1 are required for mono-ubiquitination of p50. This experiment also addresses the earlier question of whether it is BARD1/BRCA1 or a separate downstream BARD1-dependent ligase that mono-ubiquitinates p50.

5. Figure 1c, right panels: what is the high MW species that appears in both the BARD1 and IgG immunoprecipitates?

This was a non-specific band. Unfortunately, on Co-IP studies the p50 band runs very near the IgG heavy chain band (a problem not encountered with anti-BARD1 IB as BARD1 is close to 100 kDa). To address this problem, we obtained anti-BARD1 antibody conjugated to agarose beads. This corrected the problem as can be seen (**Supplementary Fig. 1b**). In addition, and in response to Reviewer #3's comments, we also preformed this experiment in primary MEFs and observed similar interaction of endogenous Bard1/p50 (**Fig 1c**).

6. Figure 1e: much of the key control band of the GFP-ER immunoprecipitate is missing. It might be better to repeat this experiment to exclude technical problems, such as poor immunoblot transfer.

As requested by the Reviewer, we repeated this experiment. The original blot in 293T cells was removed and the experiment repeated (**Supplementary Fig 1e**). The data clearly demonstrate the lack of change in the control GFP-ER samples. In addition, we performed this experiment in primary WT MEFs expressing either TopBP1-ER or GFP-ER. Again, stimulation with TAM induced the interaction of *endogenous* Bard1 and p50 only in cells expressing TopBP1-ER (**Fig 1e**).

7. In the S phase of undamaged cells, is p50-S337 phosphorylation also mediated by CHK1?

To address this question, we performed cell synchronizations studies in the presence of si-CHK1 (si-Chk1 in murine cells) using HEK293T cells and primary WT MEFs. In both cells, knockdown of CHK1 blocked p50-S337 phosphorylation during S-phase of the unperturbed cell cycle (**Supplementary Fig. 5b**). These findings indicate that p50-S337 phosphorylation is CHK1-dependent in S phase.

8. The ability of BARD1 to bind a phosphorylated p50 is very intriguing, suggesting that the BRCT domain of BARD1, like that of NBS1, has the potential to recognize both PAR and phospho-protein ligands. However, to confirm the phospho-dependent nature of the p50/BARD1 interaction, the authors should conduct two key experiments that are conventionally used to establish the validity of BRCT/phosphoprotein interactions: is the

p50/BARD1 interaction impaired by 1) phosphatase treatment and 2) by mutations of the structurally-defined BRCT phospho-recognition residues (such as BARD1- S575)?

To address this point, we performed the following experiments:

- a) We included lambda phosphatase prior to analysis of BARD1 and p50 interaction and found that this treatment blocked the interaction in both human and mouse cells (**Fig. 2h and Supplementary Fig. 2d**).
- b) We used FLAG-BARD1 constructs mutated at specific phospho-protein recognition residues based on the BARD1 BRCT crystal structure (Birrane G, et. al. 2007 *Biochem*). While S575 (in the P₁ pocket) was recommended by the Reviewer, for additional specificity controls we also mutated T617 (in P₁) and H686 (in P₂). Notably, while endogenous p50 interacted with WT BARD1, p50 did not interact with BARD1 constructs mutated at the phospho-protein recognition residues (**Fig. 2k**).

Together, loss of the BARD1/p50 interaction seen in these experiments supports the contention that the BARD1 BRCT domains specifically recognize phospho-p50.

Reviewer #3

The authors identified BARD1, a prominent BRCA1-associated protein, to interact with the p50 subunit of NF-κB based on mass spectrometric analysis of transfected 293T cells (Fig. 1). The site of interaction involved p50S238, a phosphorylation site previously implicated in maintenance of genome stability. Co-transfection assays into 293T cells were used to further characterize interactions between these proteins (Figs. 2-4). These studies implicated p50S337 and the BRCT domain of BARD1 via ATR-induced Chk1 kinase activation. The RING1 domain of BARD1 was shown to mono-ubiquitinate p50 at K354 and K356. Functional effects of p50/BARD1 interaction were tested in different ways. BARD1 interaction stabilized p50 protein levels and mono-ubiquitination resulted in cytoplasmic relocation of p50 from the nucleus (Fig. 5). Cytosolic localization was accompanied by reduced p50 binding to chromatin. Cell cycle progression of mouse embryo fibroblasts (MEFs) was affected by disrupting p50/BARD1 interactions with point mutations in either protein (Fig. 6). Introduction of WT p50, but not a mutated version that cannot be mono-ubiquitinated, reduced genomic instability in p50-deficient MEFs (Fig. 7). Finally, p50 and BARD1 levels were statistically correlated in a variety of human tumor samples (Fig. 7), suggesting that this interaction regulated genomic instability during disease.

The authors have done a comprehensive analysis of p50/BARD1 interactions (Figs. 1-4). The data is clean, however there is little quantitation of the co-IPs, GST-pull downs or phosphorylation states. Additionally, virtually all the biochemistry is carried out under over-expression conditions in co-transfected 293T cells. While this is a good place to start, additional quantitative studies utilizing endogenous proteins is required. Functional studies point to a role for p50 stabilization by BARD1 that results in accelerated cell cycle progression via cyclinE1 up-regulation. A strong point here is the comparison between 2KR and 2KR-Ub mutants for cell cycle studies. However, the effects of these mutants were not compared with

WT p50 in the same assay (Fig. 6g, h). I am also circumspect about p50 stability in the presence or absence of BARD1 which was not quantified, as well use of different cell lines to measure effects on the cell cycle.

Overall, this study identifies a plausible mechanism by which NF- κ B p50 confers genomic stability. This is an important question and I believe that the insight of a ATR-driven association of p50 with BARD1 and associated downstream effects is novel and worthy of consideration in Nature Communications. The major shortcoming in the current version of the manuscript is reliance on over-expression systems to demonstrate association and cellular effects. In my opinion, a more focused manuscript that connects DNA damage signals via endogenous p50/BARD1 to a cell cycle outcome in an untransformed cell type (e.g. MEFs) would greatly strengthen the impact of the study. Such an analysis could be added to the present data after removing much of Figs. 1-4 to supporting material. The specifics of re-organization can be discussed between the authors and editors if this option is taken.

We appreciate the comments of the Reviewer and the general critiques raised. We have identified the following specific points and have addressed each of these in the comments below.

The Reviewer asked for additional quantitative studies and the use of endogenous proteins.

This was addressed wherever possible. Specifically, we semi-quantitatively analyzed endogenous p50 S337 phosphorylation (**Supplementary Fig. 2b**) and p50 mono-ubiquitination in response to ATR activation (**Supplementary Fig. 3a**). In addition, throughout **Figures 1- 6**, we analyzed all Co-IPs, EMSAs and immunoblots using ImageJ gel band analysis as indicated in the methods. The quantification data is documented in each figure panel as the fold change in signal, normalized to loading, relative to the appropriate control lane.

We also repeated many studies to examine endogenous proteins and semi-quantitatively analyzed the change in signal where possible. This can be seen in the following figures: **Figures 1c, 1d, 1e, 2f, 2g, 2h, 6a, 6b** and **Supplementary Figures 1b, 2d, 5a, 5b and 5c**. All quantification analyses were representative of the changes seen in at least two, and most often 3, independent biological experiments.

The Reviewer noted that the effects of the p50 2KR and 2KR-Ub mutants had not been compared with WT p50 in the same assay for analysis of cell cycle studies (Figs. 6g and h).

In response to this comment, we repeated these experiments and incorporated WT-p50 and empty vector into the new studies. The results of these experiments demonstrate that p50 2KR-Ub reduces S phase cell percentage and overall cell number to a value near that seen with WT-p50. These data are presented as **Figure 6h** and **Supplementary Fig 5f**.

The Reviewer asked for quantification of p50 stability studies in presence and absence of BARD1.

This was done and the half-life ($t_{1/2}$) of the different p50 constructs determined. These data are included as **Supplementary Figs. 4a and 4b**. The data demonstrate that interaction with BARD1 stabilizes p50 and that addition of a single ubiquitin increases the $t_{1/2}$ of mutants that cannot be mono-ubiquitinated. Also, semi-quantitative analysis of the kinetics of p50 degradation in the presence of various BARD1 cancer-linked mutants was determined (**Supplementary Fig. 7b**). For the stability studies, it was necessary to use exogenous p50 because: **i)** determining the roles of the various residues required mutant p50 and **ii)** even in the presence of cycloheximide, endogenous p50 levels can be affected by the spontaneous degradation of its parental protein, p105.

The Reviewer asked for “use of different cell lines to measure effects on the cell cycle.”

This was done. Specifically, we examined the changes in p50 phosphorylation and interaction with Bard1 in primary untransformed MEFs and 293T cells (**Figs 6a and b and Supplementary Fig 5b and c**). The effect of different p50 mutants on cell proliferation is also shown using two different cell lines, MEFs and 293T cells (**Fig 6f and Supplementary Fig 5e**).

Finally, the Reviewer made a general comment noting the manuscript would benefit from better focus linking “DNA damage signals via endogenous p50/BARD1 to a cell cycle outcome in an untransformed cell type (e.g. MEFs).”

As noted above, we performed multiple additional experiments examining *endogenous* p50/BARD1 in *untransformed MEFs*. These data are included in the main and Supplementary figures (**Figures 1c, 1d, 1e, 2f, 2g, 2h, 6a, 6b and Supplementary Figure 5b**). Also, endogenous proteins were examined in other cell lines (**Supplementary Figures 1b, 2d, 5b and 5c**).

Together, the additions included address the major concerns raised by this Reviewer including: data quantification and examination of endogenous proteins in untransformed MEFs. Of course, for analysis of the roles of individual p50 residues it remained necessary to use p50 mutants. In addition, given that only a fraction of nuclear p50 (which is itself only a percentage of total p50- that is primarily cytoplasmic) is mono-ubiquitinated, analysis of this PTM required over-expression of p50 and the use of tagged ubiquitin (especially for the biochemical studies in Figs. 3 and 4).

REVIEWERS' COMMENTS

Reviewer #1 (Remarks to the Author):

Summary

The authors have addressed most of my concerns in the rebuttal and revised manuscript and only a few outstanding issues remain to be addressed or clarified.

Comment: It remains unclear about the relationship between pS328-p50 and pS337-p50 on BARD1 binding, mono-ubiquitination and genomic instability. Either p50 S328A or p50 S337A can reduce the interaction with BARD1 and induce genomic instability. However, the authors showed that loss of S328 phosphorylation by mutation did not block S337 phosphorylation in response to replication stress. I wonder whether the loss of S328 phosphorylation by mutation will affect mono-ubiquitination of p50. In addition, D mutant is usually to mimic phosphorylation. Why S337D had almost no DNA binding? The authors may need to discuss this part.

Reviewer #2 (Remarks to the Author):

In their revised manuscript, Wu et al. have clarified two important issues raised in the previous review. First, by mutational analysis and phosphatase treatment, they have convincingly shown that the interaction between p50 and the BRCT domain of BARD1 is phospho-dependent. Second, although they originally proposed that BARD1 can ubiquitinate p50 independently of BRCA1, they have now produced a compelling body of new data demonstrating that p50 ubiquitination is catalyzed by the BRCA1/BARD1 heterodimer. With these questions resolved, the authors have uncovered a novel mechanism, operative both during the normal G1-S transition and in response to exogenous replication stress, by which the NFkB and BRCA1/BARD1 pathways collaborate to ensure proper cell cycle progression and genome maintenance. This is a very important contribution, supported by clean and compelling data.

Comments:

1) Lines 55-57: As detailed in the previous review, “BRCA1-independent functions of BARD1” are likely to be minor aspects of BARD1 function – if they exist at all. In the revised manuscript, the authors adjusted the text to remove indications of BRCA1-independent actions of BARD1 and they provided compelling new evidence that p50 ubiquitination is catalyzed by the BRCA1/BARD1 heterodimer, not by BARD1 alone. However, on lines 55-57 they continue to cite the two misleading review articles from the same group (references 14 and 15) that argue for the existence of substantial BRCA1-independent BARD1 functions. Citing these reviews only perpetuates this misconception. This is unnecessary, especially as there are a number of more accurate, balanced, and informative reviews of BRCA1/BARD1 to choose from, including those of Jiang & Greenberg (J. Biol. Chem. 290: 17724-17732, 2015) and Tarsounas & Sung (Nat. Rev. Molec. Cell Biol. 21: 284–299, 2020).

2) Lines 178-179: “The only reported enzymatic activity of BARD1 is ubiquitin ligation”. It may be

preferable to adjust this statement as it might be misconstrued to imply that BARD1 possess an intrinsic E3 ligase activity (independent of BRCA1).

Response to Referees

The critiques of each Reviewer are given in black and followed by our response in red.

Reviewer #1 (Remarks to the Author):

Summary

The authors have addressed most of my concerns in the rebuttal and revised manuscript and only a few outstanding issues remain to be addressed or clarified.

Comment: It remains unclear about the relationship between pS328-p50 and pS337-p50 on BARD1 binding, mono- ubiquitination and genomic instability. Either p50 S328A or p50 S337A can reduce the interaction with BARD1 and induce genomic instability. However, the authors showed that loss of S328 phosphorylation by mutation did not block S337 phosphorylation in response to replication stress. I wonder whether the loss of S328 phosphorylation by mutation will affect mono- ubiquitination of p50. In addition, D mutant is usually to mimic phosphorylation. Why S337D had almost no DNA binding? The authors may need to discuss this part.

As suggested by the Reviewer and Editor, we addressed these findings in the discussion. First, given that our initial affinity purification and MS/MS studies suggested that p50 S328A doesn't bind BARD1, and as BARD1 binding is required for p50 mono-ubiquitination, we suspect that loss of S328 phosphorylation by mutation will also block p50 mono-ubiquitination. We have noted this in the discussion.

With respect to the S337D mutant: This mutation mimics S337 phosphorylation. Therefore, as phosphorylation of S337 by CHK1 (during S phase or in response to exogenously-induced ATR) leads to a loss of p50 chromatin recruitment, the S337D mutant would also be expected to have reduced DNA binding compared to wildtype (WT). Consistent with this, we found decreased binding of S337D on both gel shift and CHIP-Seq analysis compared to WT.

Reviewer #2 (Remarks to the Author):

In their revised manuscript, Wu et al. have clarified two important issues raised in the previous review. First, by mutational analysis and phosphatase treatment, they have convincingly shown that the interaction between p50 and the BRCT domain of BARD1 is phospho-dependent. Second, although they originally proposed that BARD1 can ubiquitinate p50 independently of BRCA1, they have now produced a compelling body of new data demonstrating that p50 ubiquitination is catalyzed by the BRCA1/BARD1 heterodimer. With these questions resolved, the authors have uncovered a novel mechanism, operative both during the normal G1-S transition and in response to exogenous replication stress, by which the NFkB and BRCA1/BARD1 pathways collaborate to ensure proper cell cycle progression and genome maintenance. This is a very important contribution, supported by clean and compelling data.

Comments:

1) Lines 55-57: As detailed in the previous review, “BRCA1-independent functions of BARD1” are likely to be minor aspects of BARD1 function – if they exist at all. In the revised manuscript, the authors adjusted the text to remove indications of BRCA1-independent actions of BARD1 and they provided compelling new evidence that p50 ubiquitination is catalyzed by the BRCA1/BARD1 heterodimer, not by BARD1 alone. However, on lines 55-57 they continue to cite the two misleading review articles from the same group (references 14 and 15) that argue for the existence of substantial BRCA1-independent BARD1 functions. Citing these reviews only perpetuates this misconception. This is unnecessary, especially as there are a number of more accurate, balanced, and informative reviews of BRCA1/BARD1 to choose from, including those of Jiang & Greenberg (J. Biol. Chem. 290: 17724-17732, 2015) and Tarsounas & Sung (Nat. Rev. Molec. Cell Biol. 21: 284–299, 2020).

In response to this comment, we have removed the two references and added the suggested new references (Refs: 16 and 17).

2) Lines 178-179: “The only reported enzymatic activity of BARD1 is ubiquitin ligation”. It may be preferable to adjust this statement as it might be misconstrued to imply that BARD1 possess an intrinsic E3 ligase activity (independent of BRCA1).

In response to this comment, we adjusted the indicated text.